# mTORC1 cooperates with tRNA wobble modification to sustain the protein synthesis machinery

Julia Hermann [1,2], Toman Borteçen[2,3], Robert Kalis [4,5], Alexander Kowar [2,6], Catarina Pechincha [1,2], Vivien Vogt[4,5], Martin Schneider[7], Dominic Helm [7], Jeroen Krijgsveld[3], Fabricio Loayza-Puch [6], Johannes Zuber [4,8] & Wilhelm Palm [1] ✉

Synthesizing the cellular proteome is a demanding process that is regulated by numerous signaling pathways and RNA modifications. How precisely these mechanisms control the protein synthesis machinery to generate specific proteome subsets remains unclear. Here, through genome-wide CRISPR screens we identify genes that enable mammalian cells to adapt to inactivation of the kinase mechanistic target of rapamycin complex 1 (mTORC1), the central driver of protein synthesis. When mTORC1 is inactive, enzymes that modify tRNAs at wobble uridines ($U_{34}$-enzymes), Elongator and Ctu1/2, become critically essential for cell growth in vitro and in tumors. By integrating quantitative nascent proteomics, steady-state proteomics and ribosome profiling, we demonstrate that the loss of $U_{34}$-enzymes particularly impairs the synthesis of ribosomal proteins. However, when mTORC1 is active, this biosynthetic defect only mildly affects steady-state protein abundance. By contrast, simultaneous suppression of mTORC1 and $U_{34}$-enzymes depletes cells of ribosomal proteins, globally inhibiting translation. Thus, mTORC1 cooperates with tRNA $U_{34}$-enzymes to sustain the protein synthesis machinery and support the high translational requirements of cell growth.

Proteins constitute more than 50% dry mass of mammalian cells, and their production consumes large amounts of biosynthetic and bioenergetic resources[1,2]. Even the protein synthesis machinery itself—comprising ribosomes, tRNAs, mRNAs and various translation factors—makes up a substantial fraction of cellular mass. In fact, the about 80 ribosomal proteins alone account for 4–6% of total cellular protein[3,4]. The abundance and activity of the protein synthesis machinery are thus carefully regulated by numerous signaling pathways, covalent modifications and transcriptional processes to ensure that the output of protein synthesis matches cellular demands[5].

The kinase mechanistic target of rapamycin complex 1 (mTORC1) is a central regulator of protein synthesis and cell growth[6,7]. mTORC1 is activated by concerted inputs from growth factors and nutrients. In turn, mTORC1 phosphorylates multiple targets to enhance the initiation and elongation steps of mRNA translation, thereby rapidly stimulating the production of new proteins. mRNAs containing

[1]Division of Cell Signaling and Metabolism, German Cancer Research Center (DKFZ) and DKFZ-ZMBH Alliance, Heidelberg, Germany. [2]Faculty of Biosciences, University of Heidelberg, Heidelberg, Germany. [3]Division of Proteomics of Stem Cells and Cancer, German Cancer Research Center (DKFZ), Heidelberg, Germany. [4]Research Institute of Molecular Pathology (IMP), Vienna BioCenter (VBC), Vienna, Austria. [5]Vienna BioCenter PhD Program, Doctoral School of the University of Vienna and Medical University of Vienna, Vienna BioCenter (VBC), Vienna, Austria. [6]Translational Control and Metabolism Group, German Cancer Research Center (DKFZ), Heidelberg, Germany. [7]Proteomics Core Facility, German Cancer Research Center (DKFZ), Heidelberg, Germany. [8]Medical University of Vienna, Vienna BioCenter (VBC), Vienna, Austria. ✉e-mail: w.palm@dkfz-heidelberg.de

5′ terminal oligopyrimidine (TOP) motifs, which encode ribosomal proteins and several translation factors[8,9], are particularly sensitive to translational up-regulation by mTORC1[8,10,11]. By increasing the production of the components necessary for building the protein synthesis machinery, mTORC1 signaling thus enhances the translational capacity of cells over time. While mechanisms through which mTORC1 promotes mRNA translation are well-understood, it remains unclear, how inputs from mTORC1 and other regulators of protein synthesis are integrated to control translational output.

During protein synthesis, tRNAs function as adaptors that translate the genetic code of mRNA codons into the corresponding amino acids. tRNA molecules are decorated with various chemical modifications that enhance the efficiency and accuracy of translation[12,13]. Uridines in the anticodon wobble position ($U_{34}$) of tRNAs are modified by two sets of enzymes ($U_{34}$-enzymes): the Elongator complex (consisting of Elp1-Elp6) initiates the formation of methoxycarbonylmethyl ($mcm^5$) groups; and Cytosolic Thiouridylase Subunits 1/2 (Ctu1-Ctu2) add a thiol ($s^2$) group to produce $mcm^5s^2U_{34}$[14-17]. Wobble uridine tRNA modifications enhance the decoding mainly of three codons, AAA (lysine), CAA (glutamine) and GAA (glutamate)[18-21], hereafter referred to as VAA codons (UAA is a stop codon and thus not translated).

tRNA $U_{34}$ modification is implicated in various physiological and pathological processes[16,22]. Yeast lacking $U_{34}$-enzymes display increased sensitivity to environmental stresses, which results from reduced translation of VAA codon-rich stress response genes and increased proteotoxic stress[20,23-25]. In mammals, mutations in $U_{34}$-enzymes lead to neurological disorders through induction of the unfolded protein response[18,26]. Several cancers depend on $U_{34}$-enzymes to efficiently express oncoproteins that are encoded by VAA codon-rich mRNAs[19,27,28]. Whether $U_{34}$-enzymes functionally interact with other translational regulators, and how precisely tRNA wobble uridine modification enhances the production of specific proteome subsets remain outstanding questions.

Here, through genome-wide CRISPR screens in mammalian cells we identified a functional interaction between mTORC1 and tRNA $U_{34}$-modifying enzymes in sustaining protein synthesis and cell growth. By integrating nascent proteomics, steady-state proteomics and ribosome profiling, we systematically analyzed how mTORC1 and $U_{34}$-enzymes shape the cellular proteome. $U_{34}$-enzymes were globally required for efficient protein synthesis and especially enhanced the production of ribosomal proteins in a codon-dependent manner, but loss of $U_{34}$-enzymes had only a slight impact on steady-state ribosomal protein abundance when mTORC1 was active. However, concerted suppression of mTORC1 and $U_{34}$-enzymes depleted cells of ribosomal proteins, strongly decreasing protein synthesis and inhibiting cell proliferation in vitro and in tumors. Thus, cooperation of mTORC1 with $U_{34}$-enzymes promotes the generation of the protein synthesis machinery to enhance translational capacity and support the biosynthetic demands of cell growth.

## Results

### A CRISPR screen identifies tRNA wobble uridine-modifying enzymes as essential for cell growth during mTOR inhibition

To systematically identify genes that sustain cell growth during mTOR inhibition, we conducted a proliferation-based CRISPR screen in cells that were treated with the mTOR kinase inhibitor, torin 1. As primary screening model, we used a single cell-derived clone of the murine pancreatic cancer cell line EPP2, which was engineered to express doxycycline-inducible Cas9 (iCas9) for time-controlled gene editing (Suppl. Fig. 1a). Following transduction of the Vienna genome-wide sgRNA library[29], drug selection and doxycycline-induced CRISPR/Cas9 mutagenesis, cells were cultured for 14 population doublings in the presence of torin 1 or solvent control. Throughout the screen, torin 1 decreased population doubling times by ~70% (Fig. 1a) and suppressed mTOR signaling (Suppl. Fig. 1b). We then used deep sequencing and

MAGeCK[30] to quantify changes in sgRNA representation between torin 1-treated and control cell populations (Fig. 1b and Suppl. Datas 1 and 2) and between either condition and the starting population (Suppl. Fig. 1c, d). About 190 genes scored as selectively essential during torin 1 treatment, including components of mTORC1 signaling and protein synthesis. Conversely, about 80 genes scored as selectively nonessential during torin 1 treatment, including various signaling proteins and negative mTORC1 regulators. Selected screen hits were validated using competitive proliferation assays (Suppl. Fig. 1e).

One prominent group of genes that we identified as selectively essential during mTOR inhibition was a class of tRNA-modifying enzymes which generate $mcm^5s^2$ modifications at anticodon wobble uridines ($U_{34}$-enzymes) (Fig. 1b, c and Suppl. Fig. 1c, d). These included several components of the Elongator complex (Elp2, Elp3, Elp6) and Cytosolic Thiouridylase Subunits 1/2 (Ctu1/2)[14-17]. Kti12, an accessory regulator of Elongator[31], and Mocs3, which mobilizes sulfur for the tRNA thiolation reaction[32], were also more essential during torin 1 treatment. Most other tRNA-modifying enzymes did not display differential knockout phenotypes between torin 1 and control conditions (Suppl. Fig. 1c, d).

To confirm that $U_{34}$-enzymes are required for cell proliferation in the context of mTOR inhibition, we generated Ctu1 and Elp3 inducible knockout (iKO) EPP2 cells (Suppl. Fig. 2a, b). By itself, loss of Ctu1 or Elp3 led to a moderate decrease in cell proliferation (Fig. 1d, e); cell size and cellular protein content were not significantly affected (Suppl. Fig. 2c, d). The knockout cells displayed increased sensitivity to torin 1 over a range of inhibitor concentrations (Suppl. Fig. 2e), despite retaining higher mTORC1 activity as compared to control cells (Suppl. Fig. 2f, g). At a low concentration, torin 1 selectively and effectively suppressed the proliferation of Ctu- or Elp3-deficient cells (Fig. 1d, e). Similarly, loss of Ctu1 or Elp3 rendered cells highly sensitive to rapamycin, a partial mTORC1 inhibitor that had minimal impact on the proliferation of control cells (Fig. 1d, e and Suppl. Fig. 2h). Ctu1/Elp3 double knockout (iDKO) cells had more pronounced growth defects, consistent with largely non-redundant $U_{34}$-enzyme functions (Suppl. Fig. 2i). We further generated Ctu1 iKOs in various cancer cell lines and in murine embryonic fibroblasts (MEFs), as well as Ctu1 knockout (KO) clones derived from parental cells (Suppl. Fig. 2j). Although the sensitivity of the different cell lines to torin 1 and rapamycin displayed some variation, their growth was strongly and consistently inhibited by co-suppression of mTORC1 and Ctu1 (Suppl. Fig. 2k–m). The proliferation and rapamycin sensitivity of Ctu1- or Elp3-deficient cells was rescued by ectopic expression of the corresponding cDNAs, confirming specificity of CRISPR/Cas9 editing (Fig. 1f–h).

mTOR kinase is part of two protein complexes, mTORC1 and mTORC2. While torin 1 inhibits both complexes, rapamycin is selective for mTORC1 (Suppl. Fig. 2h)[33], suggesting that $U_{34}$-enzyme-deficient cells are dependent specifically on mTORC1. To address the relevance of the two mTOR complexes directly, we transduced cells with sgRNAs targeting the mTORC1 component Raptor or the mTORC2 component Rictor. Loss of Raptor strongly suppressed the proliferation of Ctu1 KO cells, while control cells were less affected (Fig. 2a and Suppl. Fig. 3a). By contrast, loss of Rictor did not have an effect on the proliferation or rapamycin sensitivity of Ctu1 KO cells (Fig. 2b and Suppl. Fig. 3b). Next, we examined the PI3-kinase-Akt-Rheb axis, which communicates growth factor signals to mTORC1[6,7]. The proliferation of Ctu1-deficient cells was selectively suppressed by genetic ablation of Rheb (Fig. 2c and Suppl. Fig. 3c) and by pharmacological inhibition of PI3-kinase or Akt (Fig. 2d and Suppl. Fig. 3d), albeit to a lesser extent than by inhibition of mTORC1. By contrast, Ctu1-deficient and control cells displayed comparable sensitivity to other growth-suppressive perturbations, including MAPK pathway inhibitors (Suppl. Fig. 3e, f) and osmotic stress (Suppl. Fig. 3g). Thus, mammalian cells lacking $U_{34}$-enzymes are not sensitive to growth inhibition in general, but specifically to suppression of the mTORC1 signaling pathway.

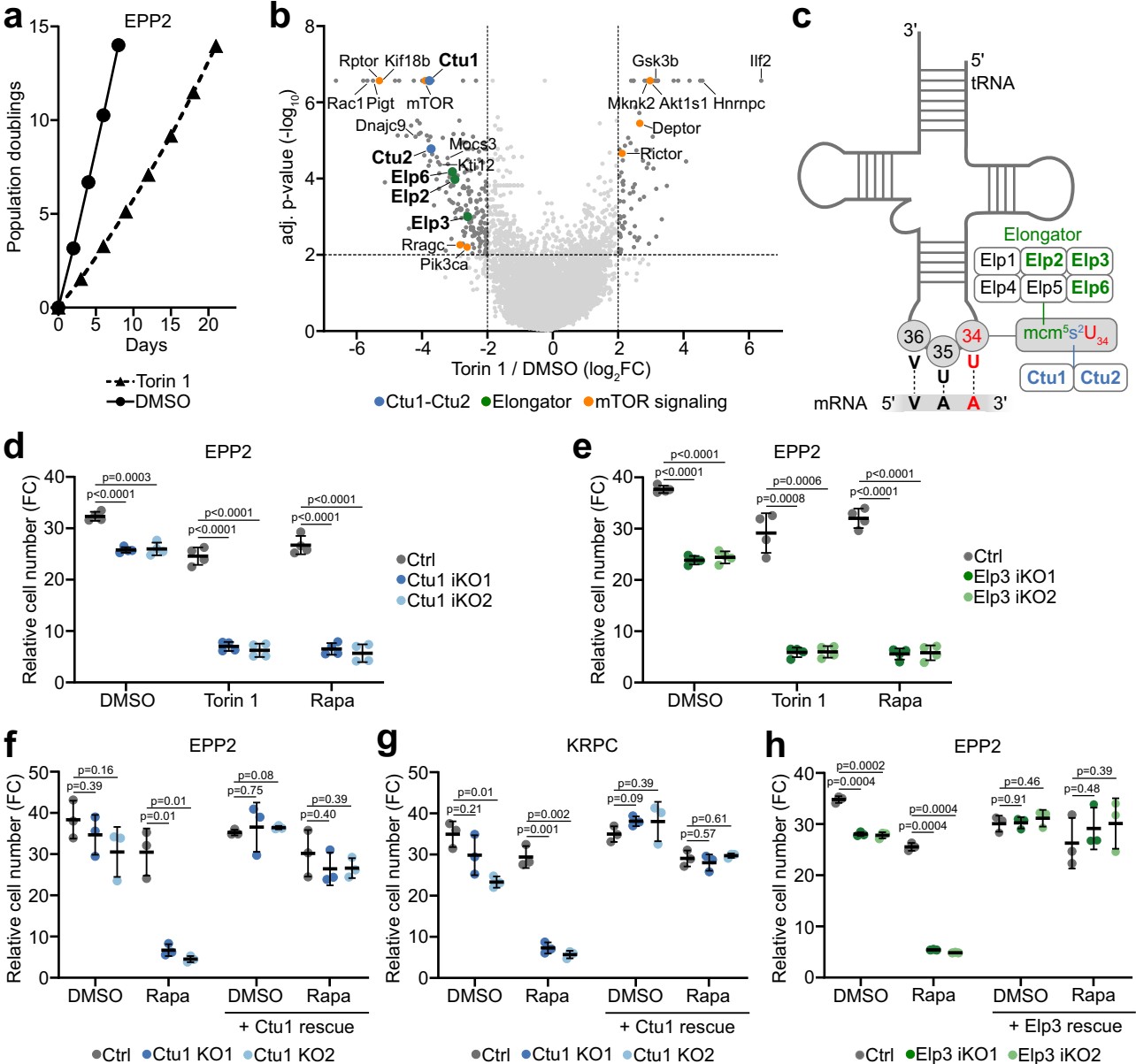

**Fig. 1 | A CRISPR screen identifies tRNA wobble enzymes as essential for cell proliferation during mTOR inhibition. a** Population doublings of EPP2 iCas9 cells during the CRISPR screen in the presence of torin 1 [300 nM] or dimethyl sulfoxide (DMSO). **b** Gene-level enrichment or depletion of sgRNAs in the CRISPR screen upon torin 1 treatment. Dashed lines demarcate genes that are significantly depleted or enriched ($\log_2$ fold change ($\log_2$FC) > |2| , adj. $p$ < 0.01). Selected hits are highlighted. **c** Schematic representation of tRNA wobble uridine ($U_{34}$) modification by the Elongator complex (Elp1-Elp6) and Cytosolic Thiouridylase (Ctu1-Ctu2). Enzyme subunits that scored as selectively essential during torin 1 treatment in the screen are highlighted. **d**, **e** Fold change (FC) in cell number of (**d**) Ctu1 iKO and (**e**) Elp3 iKO EPP2 cells after 3 days in culture ± torin 1 [25 nM] or rapamycin [12.5 nM]. **f**, **g** Fold change in cell number of Ctu1 KO f) EPP2 cells and (**g**) KRPC cells with ectopic expression of Ctu1 cDNA after 3 days ± rapamycin [12.5 nM]. **h** Fold change in cell number of Elp3 iKO EPP2 cells with ectopic expression of Elp3 cDNA after 3 days ± rapamycin [12.5 nM]. **d**–**h** Data are represented as replicate mean ± SD (**d**, **e**) $n$ = 4, **f**–**h** $n$ = 3 independent experiments in 3 technical replicates); $p$ values were calculated by two-tailed unpaired $t$-test with Welch correction. Source data are provided as a Source Data file.

Finally, we investigated whether the loss of $U_{34}$-enzymes sensitizes cancer cells to mTORC1 inhibition in vivo using an orthotopic transplantation model of pancreatic cancer. After establishment of pancreatic tumors derived from Ctu1-deficient or control EPP2 cells, animals were treated with rapamycin or vehicle for 9 days. Individually, these interventions showed limited efficacy: Tumor growth was not affected by genetic ablation of Ctu1 (Fig. 2e and Suppl. Fig. 3h) and only slightly decreased by rapamycin. By contrast, tumor growth was potently suppressed by the combination of Ctu1 deletion and rapamycin treatment. Taken together, these results reveal that the loss of $U_{34}$-enzymes renders cells highly dependent on the mTORC1 signaling pathway in vitro and in tumors.

## Loss of tRNA wobble enzymes leads to a global decrease in protein synthesis

The mTORC1 inhibitor sensitivity of cells lacking tRNA wobble $U_{34}$-modifying enzymes raised the question whether mTORC1 functionally interacts with tRNA wobble uridine modification in the regulation of protein synthesis. Because the precise impact of $U_{34}$-enzymes on protein synthesis is not well defined, to address this we first investigated how $U_{34}$-enzymes affect translational output using quantitative analysis of the newly synthesized proteome (QuaNPA)[34]. Nascent proteins were labeled through a combined pulse with the methionine analog azidohomoalanine (AHA) and SILAC in Ctu1- or Elp3-deficient cells and controls. AHA-labeled proteins were enriched from

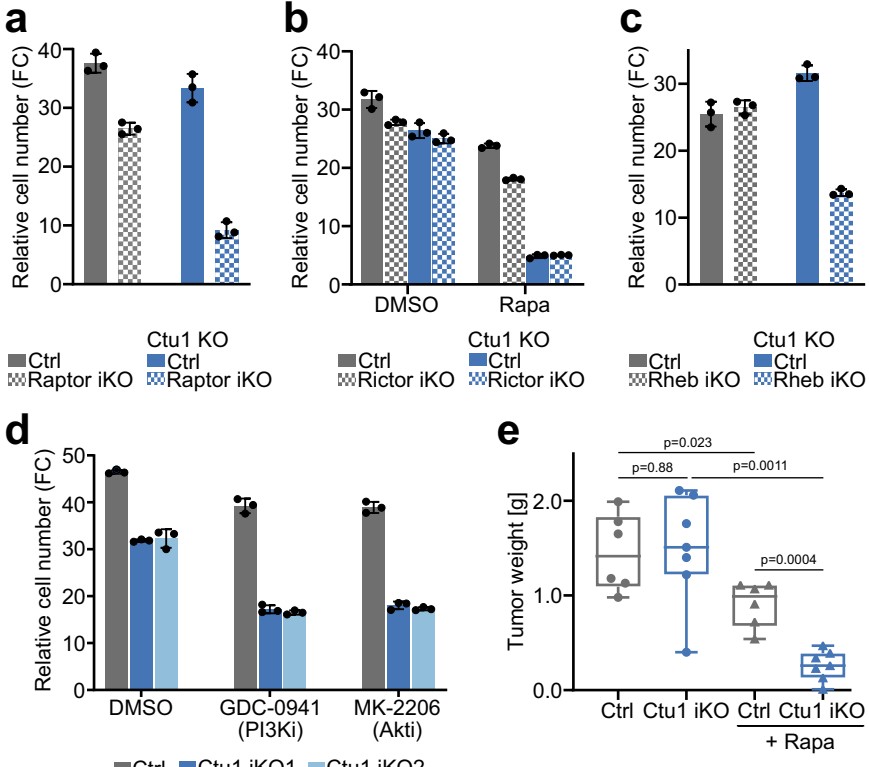

**Fig. 2 | Loss of tRNA wobble modification sensitizes cancer cells to mTORC1 pathway inhibition in vitro and in tumors. a–c** Fold change in cell number of single cell-derived clonal Ctu1 KO or control EPP2 cells expressing sgRNAs against (**a**) Raptor, (**b**) Rictor ± rapamycin [12.5 nM] and (**c**) Rheb after 3 days in culture. **d** Fold change in cell number of Ctu1 iKO EPP2 cells after 3 days in culture ± GDC-0941 [1 μM] or MK-2206 [2 μM]. **a–d** Data are represented as mean ± SD (n = 3 technical replicates). **e** Weight of orthotopic pancreatic tumors from Ctu1 iKO or control EPP2 cells in C57BL/6 J Rag2$^{-/-}$ mice after 9 days treatment with rapamycin [5 mg/kg/day] or vehicle. Centre line represents median, upper and lower bounds of the box 75th and 25th percentiles, whiskers min to max (control ± rapamycin n = 6, Ctu1 iKO ± rapamycin n = 7); p values were calculated by unpaired two-tailed t-test with Welch correction. Source data are provided as a Source Data file.

whole-cell lysates using click chemistry. Changes in protein abundance were quantified in AHA-enriched samples and in whole-cell lysates through liquid chromatography–mass spectrometry and SILAC analysis. Normalizing the newly synthesized proteome data with the median SILAC ratios of whole-cell lysates then allowed us to accurately determine the impact of U$_{34}$-enzymes on protein synthesis. About 90% of all detected proteins were decreased in the newly synthesized proteome of Ctu1-deficient cells (Fig. 3a and Suppl. Data 3). Similarly, loss of Elp3 led to a global decrease of the newly synthesized proteome (Fig. 3b). Thus, tRNA U$_{34}$-modifying enzymes play a critical role in sustaining protein synthesis.

To further investigate the impact of U$_{34}$-enzymes on translational output, we next confirmed their relevance for the translation of specific codons using ribosome profiling and differential ribosome codon reading (diricore) analysis[35]. Loss of Ctu1 led to a strong increase in ribosomal A site occupancy at AAA codons and to a lesser extent at CAA/GAA codons (Fig. 3c). Ribosomal occupancy was also significantly increased at the arginine codon AGA, consistent with previous observations[18,20]. At other codons, ribosomal occupancy was not perturbed. Next, we examined how mRNA codon content corresponds to protein synthesis changes in cells lacking U$_{34}$-enzymes. Increasing VAA codon content of mRNAs was correlated with decreasing abundance of the corresponding proteins in the newly synthesized proteome of Ctu1- or Elp3-deficient cells (Fig. 3d, e and Suppl. Fig. 4a). At the level of individual codons, this bias was evident in particular for AAA and GAA. The reverse trend was observed for proteins that were efficiently synthesized in Ctu1- or Elp3-deficient cells; their mRNAs were overall enriched in non-VAA codons, whose translation does not depend on U$_{34}$-enzymes (Suppl. Fig. 4b). Thus, U$_{34}$-enzymes are required for

efficient synthesis of proteins from mRNAs that are rich in VAA codons, consistent with previous results[17–20].

## tRNA wobble modification promotes ribosomal protein synthesis

To identify proteins whose synthesis is particularly dependent on U$_{34}$-enzymes, we performed gene set enrichment analysis (GSEA) using the nascent proteomics datasets. Ribosomal proteins were the most decreased class of proteins (Fig. 4a). Indeed, loss of either Ctu1 or Elp3 led to a global, comparable decrease of ribosomal proteins in the newly synthesized proteome, without reducing their transcript levels (Fig. 4b, c; Suppl. Fig. 5a and Suppl. Datas 3 and 4). To understand why ribosomal proteins depend on U$_{34}$-enzymes, we examined their mRNAs. In Ctu1-deficient cells, ribosomal protein mRNAs showed a pronounced increase in ribosomal occupancy at AAA codons and to a lesser extent at CAA/GAA codons (Fig. 4d). By contrast, the synonymous U$_{34}$-enzyme-independent codons were normally translated. Most ribosomal protein mRNAs are highly enriched in AAA, which encodes for lysine, a positively charged amino acid that participates in protein-RNA interactions within the ribosome (Suppl. Fig. 5b). Consistently, high mRNA AAA content and decreased synthesis of ribosomal proteins were correlated in Ctu1- or Elp3-deficient cells (Fig. 4e; Suppl. Fig. 5c and Suppl. Data 5). Of note, ribosomal protein mRNAs are not overall enriched in VAA codons as their CAA/GAA codon content is comparably low (Fig. 4f and Suppl. Fig. 5b, c). This suggests that the efficient translation of ribosomal protein mRNAs depends on U$_{34}$-enzymes due to their high AAA codon content.

In addition to ribosomal proteins, protein groups related to the biogenesis of ribosomes and other ribonucleoprotein complexes,

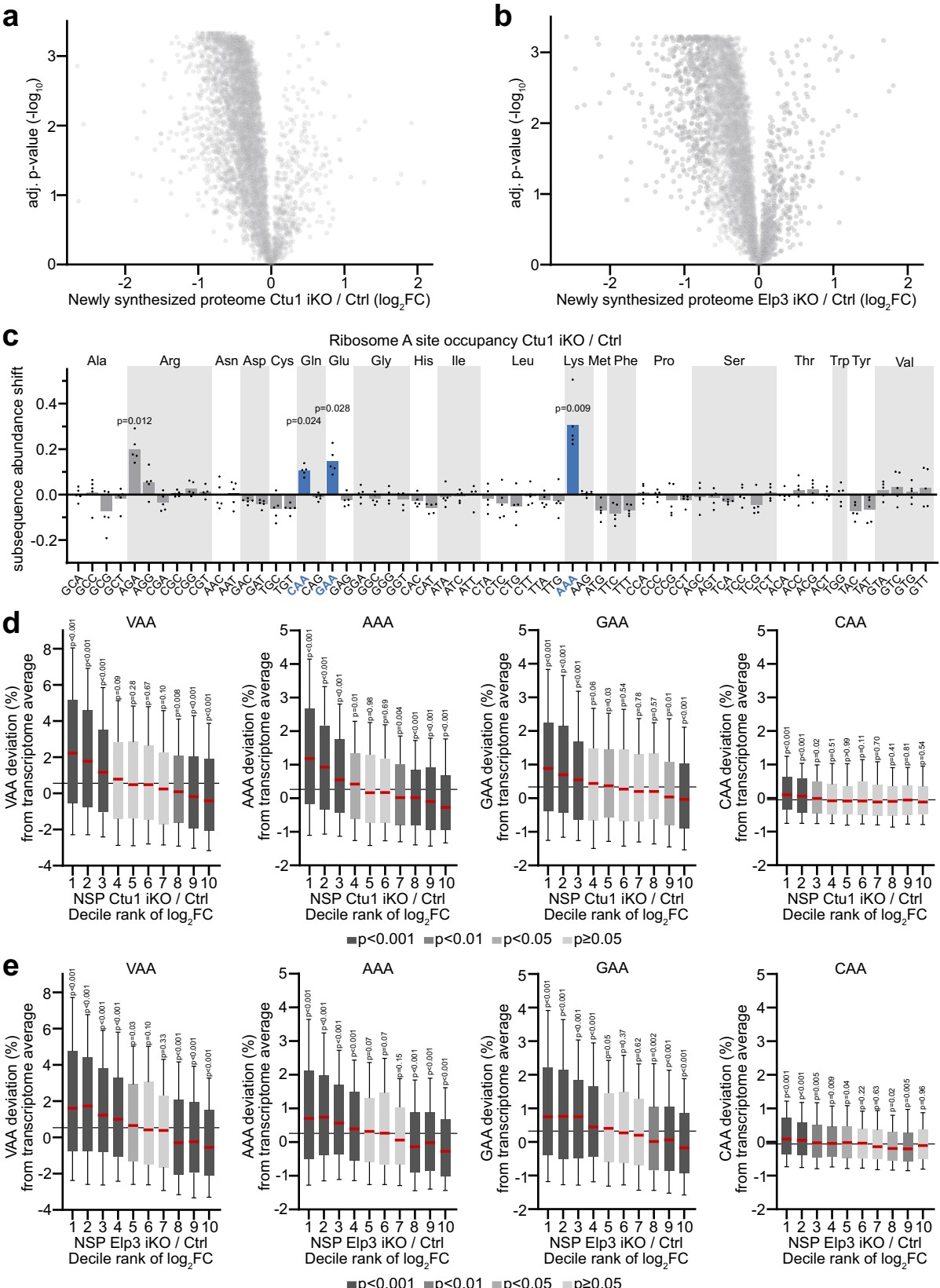

**c** Ribosome A site occupancy Ctu1 iKO / Ctrl

**d**

**e**

■ p<0.001  ■ p<0.01  ■ p<0.05  ■ p≥0.05

non-coding RNA processing, the G2 DNA damage checkpoint and sister chromatid segregation were particularly depleted in the newly synthesized proteome of U34-enzyme-deficient cells (Fig. 4a). The decreased synthesis of the different protein groups correlated with an enrichment of mRNA VAA codons, specifically AAA and GAA (Suppl.

Fig. 5c and Suppl. Data 5). Kinesins were also depleted, consistent with a previous study showing that their VAA codon-rich mRNAs depend on tRNA U34 modification[36]. In contrast to the majority of proteins, collagens and other extracellular matrix proteins were increased in the newly synthesized proteome of U34-enzyme-deficient cells (Fig. 4a).

**Fig. 3 | Loss of tRNA wobble enzymes leads to a global decrease in protein synthesis. a, b** Changes in protein synthesis of (**a**) Ctu1 iKO and (**b**) Elp3 iKO EPP2 cells, measured using the mass spectrometry (MS) based quantitative analysis of the newly synthesized proteome (QuaNPA) workflow ($n = 3$ experimental replicates). **c** Codon-specific changes in ribosome A site occupancy in Ctu1 iKO EPP2 cells, analyzed by diricore. VAA codons (i.e., AAA/CAA/GAA), which are recognized by $U_{34}$ wobble tRNAs, are highlighted. Data are represented as mean ($n = 5$ independent experiments); $p$ values were calculated by out-of-frame analysis and are shown for significant increases. **d, e** Correlation of mRNA codon usage deviation and change in the newly synthesized proteome (NSP) of (**d**) Ctu1 iKO and (**e**) Elp3 iKO EPP2 cells ($n = 3$ experimental replicates). Newly synthesized proteome data are represented as decile ranks of $\log_2 FC$ (i.e., 1 = 10 % most decreased proteins in iKO/Ctrl). In box plots, centre line represents median, upper and lower bounds of the box 75th and 25th percentiles, whiskers 10th to 90th percentiles; $p$ values were calculated by two-tailed Wilcoxon rank test. Grey line represents median codon usage of all quantified proteins. Source data are provided as a Source Data file.

Extracellular matrix protein mRNAs are not enriched in VAA codons, which may explain why their translation is resistant to the loss of Ctu1 or Elp3 (Suppl. Fig. 5c). Taken together, these results identify protein groups whose efficient synthesis depends on tRNA $U_{34}$-modifying enzymes, including most notably ribosomal proteins.

To determine how the perturbed protein synthesis in $U_{34}$-enzyme-deficient cells affects steady-state protein levels, we analyzed their proteome using liquid chromatography–mass spectrometry and label-free quantification. Proteins that decreased in the proteome of Ctu1- or Elp3-deficient cells showed a progressive increase in mRNA VAA codon content (Fig. 4g, h). GSEA identified proteins related to RNA processing, RNA metabolism, biogenesis of ribosomes and other ribonucleoprotein complexes as the most significantly decreased protein groups (Fig. 4i). Consistently, the different $U_{34}$-enzyme-dependent protein groups showed corresponding decreases in the newly synthesized and steady-state proteomes of Ctu1- or Elp3-deficient cells (Suppl. Fig. 6 and Suppl. Datas 3 and 6). However, despite the pronounced decrease of ribosomal proteins in the newly synthesized proteome, their steady-state levels were less affected (Fig. 4j, k and Suppl. Fig. 7). Thus, the translational defects resulting from loss of $U_{34}$-enzymes only partially materialize at the proteome level.

## mTORC1 and tRNA wobble enzymes converge on promoting ribosomal protein synthesis

The above results establish a role for $U_{34}$-enzymes in ribosomal protein synthesis. Intriguingly, increased production of ribosomal proteins is also a key translational program downstream of mTORC1[10,11]. To examine whether mTORC1 and $U_{34}$-enzymes functionally interact in ribosomal protein synthesis, we first determined the relevance of mTORC1 for codon-specific translation. Short treatment with mTORC1 inhibitors did not perturb ribosomal codon occupancy in control cells (Fig. 5a), affect the accumulation of ribosomes at VAA codons in Ctu1-deficient cells (Fig. 5b), or aggravate the overall decreased protein synthesis from VAA-rich mRNAs in Ctu1-deficient cells (Suppl. Fig. 8a, b). Individually, rapamycin or a moderate dose of torin 1 also did not affect global protein synthesis (Fig. 5c, d) including the synthesis of most $U_{34}$-enzyme-dependent protein groups (Fig. 5e, f and Suppl. Fig. 8c–e). However, ribosomal proteins were strongly depleted in the newly synthesized proteome upon mTORC1 inhibition (Fig. 5c–f and Suppl. Data 3). Thus, mTORC1 is required for ribosomal protein synthesis but not for the efficient translation of VAA codons.

Next, we investigated the functional interaction of mTORC1 and $U_{34}$-enzymes. Nascent proteomics revealed a global decrease of protein synthesis in Ctu1-deficient cells treated with torin 1 or rapamycin as compared to the individual manipulations, with ribosomal proteins being particularly strongly decreased (Fig. 6a, b and Suppl. Fig. 8f, g). Thus mTORC1 and $U_{34}$-enzymes converge on promoting the production of ribosomal proteins. To determine whether ribosomal proteins are direct targets of $U_{34}$-enzymes, we generated several ribosomal protein variants in which mRNA VAA codons were replaced by synonymous, $U_{34}$-enzyme-independent VAG codons. Expression constructs further contained a TOP motif for translational regulation by mTORC1 and an HA-tag for quantitative immunoblotting. We then examined expression of the ribosomal protein variants in Ctu1-deficient cells treated with rapamycin and, as excess ribosomal proteins are rapidly degraded by the proteasome[37], with proteasome inhibitor. Loss of Ctu1 decreased the expression of wild-type Rpl29 and Rps25, consistent with their high mRNA VAA codon content (Fig. 6c–f). By contrast, the VAG mutant variants were efficiently expressed in Ctu1-deficient cells, at the same level as in control cells. Thus, Rpl29 and Rps25 require Ctu1 for codon-dependent mRNA translation. We also examined Rps3, whose mRNA has an unusually low VAA codon content for a ribosomal protein. Indeed, wild-type and VAG mutant Rps3 variants were expressed at comparable levels, independently of Ctu1 (Fig. 6g, h).

To determine the collective impact of mTORC1 signaling and $U_{34}$-enzymes on the steady-state levels of ribosomal proteins, we measured the proteome of Ctu1- or Elp3-deficient cells after long-term mTORC1 inhibition with rapamycin. Individually, loss of Ctu1 or Elp3 and rapamycin treatment only slightly decreased ribosomal protein levels (Fig. 7a, b and Suppl. Data 6). However, ribosomal proteins were strongly decreased by the combination of Ctu1 or Elp3 deficiency and rapamycin (Fig. 7c). Quantitative immunoblotting confirmed that ribosomal protein Rpl29 was depleted in Ctu1- or Elp3-deficient cells treated with mTORC1 inhibitors (Fig. 7d, e), which was rescued by ectopic expression of Ctu1 or Elp3 cDNA (Suppl. Fig. 9a, b). Similarly, co-suppression of Ctu1/Elp3 and mTORC1 strongly reduced Rpl29 levels in various human and mouse cell lines (Fig. 7f–m and Suppl. Fig. 9c, d). Interestingly, Rps3 was also decreased, although its mRNA is not enriched in VAA codons and does not require $U_{34}$-enzymes for efficient translation (Fig. 6g, h), conceivably as a secondary consequence of the decreased synthesis of other ribosomal proteins. Besides suppressing protein synthesis, mTORC1 inhibition also increases autophagy, a catabolic process during which ribosomes and other cytosolic constituents are degraded in lysosomes[38]. However, basal autophagic flux was not affected in Ctu1- or Elp3-deficient cells (Suppl. Fig. 9e), and genetic ablation of autophagy did not prevent the decrease of ribosomal proteins upon rapamycin treatment (Suppl. Fig. 9f). Thus, mTORC1 inhibition does not deplete ribosomal proteins in $U_{34}$-enzyme-deficient cells through induction of autophagy.

## mTORC1 cooperates with tRNA wobble enzymes to sustain translational capacity

The above results demonstrate that mTORC1 and $U_{34}$-enzymes converge on promoting the production of ribosomal proteins. To determine the impact on translational capacity, we subjected $U_{34}$-enzyme-deficient cells to long-term treatment with mTORC1 inhibitors to deplete ribosomal proteins. We then measured puromycin incorporation into nascent polypeptides using quantitative immunoblotting as a proxy for global translational output. Loss of Ctu1 caused a slight decrease in puromycin incorporation (Fig. 8a, b and Suppl. Fig. 10a), consistent with the decreased protein synthesis that was observed by nascent proteomics (Fig. 3a). Long-term torin 1 treatment of control cells also led to a moderate decrease in puromycin incorporation. However, torin 1 combined with Ctu1 deletion led to a strong suppression of puromycin incorporation (Fig. 8a, b and Suppl. Fig. 10a). Similarly, puromycin incorporation was potently suppressed in Ctu1-deficient cells treated with rapamycin and inhibitors against PI3-kinase or Akt (Fig. 8c and Suppl. Fig. 10b), as well as in Elp3-deficient or Ctu1/Elp3 double-deficient cells treated with torin 1 (Fig. 8d–g and Suppl. Fig. 10c, d). Thus, long-term inhibition of mTORC1 or upstream growth factor signaling synergizes with genetic ablation of tRNA $U_{34}$-modifying enzymes to suppress global protein synthesis.

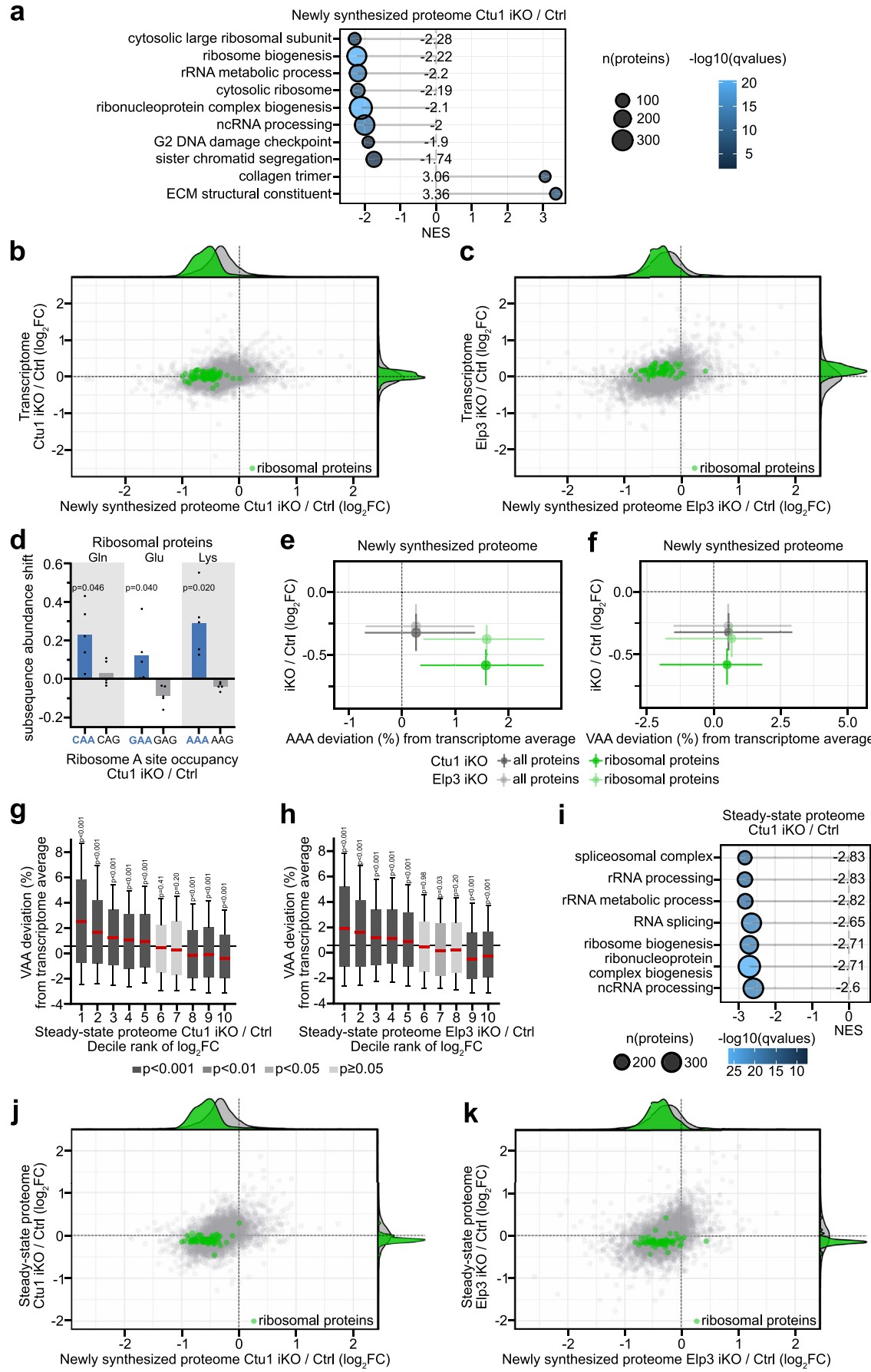

**Fig. 4 | tRNA wobble modification promotes ribosomal protein synthesis.**
**a** GSEA showing highly depleted or enriched protein groups in the newly synthesize proteome of Ctu1 iKO EPP2 cells. NES, normalized enrichment score. **b, c** Scatter plot comparing changes in the newly synthesized proteome versus transcriptome of (**b**) Ctu1 iKO and (**c**) Elp3 iKO EPP2 cells. Transcript levels were quantified by RNA sequencing ($n = 2$ independent experiments). **d** Changes in ribosome A site occupancy of $U_{34}$-wobble enzyme-dependent/independent VAA/VAG codons on ribosomal protein mRNAs in Ctu1 iKO EPP2 cells, analyzed by diricore. Data are represented as mean ($n = 5$ independent experiments); $p$ values were calculated by out-of-frame analysis and are shown for significant increases. **e, f** Correlation of mRNA codon usage deviation for (**e**) AAA and (**f**) VAA with changes in the newly synthesized proteome of Ctu1 iKO and Elp3 iKO EPP2 cells. Data are represented as

median and interquartile range. **g, h** Correlation of mRNA codon usage deviation and change in the steady-state proteome of (**g**) Ctu1 iKO and (**h**) Elp3 iKO EPP2 cells. Steady-state proteomes were quantified by label-free mass spectrometry ($n = 5$ independent experiments); proteome data are represented as decile ranks of $log_2FC$ (i.e., 1 = 10% most decreased proteins in iKO / Ctrl). In box plots, centre line represents median, upper and lower bounds of the box 75th and 25th percentiles, whiskers 10th to 90th percentiles; $p$ values were calculated by two-tailed Wilcoxon rank test. Grey line represents median codon usage of all quantified proteins. **i** GSEA showing the top depleted protein groups in the steady-state proteome of Ctu1 iKO EPP2 cells. **j, k** Scatter plot comparing changes in the newly synthesized proteome versus steady-state proteome of (**j**) Ctu1 iKO and (**k**) Elp3 iKO EPP2 cells. Source data are provided as a Source Data file.

Finally, we dissected contributions of translation-promoting signaling and ribosomal protein production in sustaining protein synthesis. First, we determined the time frame over which mTORC1 inhibition acts. Although short-term torin 1 treatment potently inhibited mTORC1 signaling in Elp3-deficient cells, puromycin incorporation was only moderately decreased (Fig. 8h). By contrast, persistent torin 1 treatment almost completely suppressed puromycin incorporation in Elp3-deficient cells. To uncouple the role of mTORC1 in translation-promoting signaling and in sustaining ribosomal biogenesis, we treated Elp3-deficient cells with torin 1 for 40 h and acutely reactivated mTORC1 by washout of torin 1. In control cells, removal of torin 1 led to a rapid activation of mTORC1 signaling and a concomitant increase in puromycin incorporation (Fig. 8i, j). However, although Elp3-deficient cells reactivated mTORC1 to a similar extent, they displayed a blunted increase in puromycin incorporation. This suggests that simultaneous suppression of mTORC1 and tRNA $U_{34}$-modifying enzymes limits the translational capacity of cells by depleting the protein synthesis machinery.

## Discussion

The present study establishes a functional interaction between mTORC1 signaling and a tRNA modification pathway in protein synthesis. By systematically analyzing how mTORC1 and tRNA $U_{34}$-enzymes shape the cellular proteome, we identify the synthesis of ribosomal proteins as a point of convergence of these translational regulators. Concerted suppression of mTORC1 and $U_{34}$-enzymes depletes cells of ribosomal proteins, leading to the collapse of translational capacity and the inhibition of cell growth in vitro and in orthotopic tumors. Thus, mTORC1 cooperates with tRNA wobble uridine modification to tune the protein synthesis machinery for supporting the high translational demands of rapidly proliferating cells.

Ribosome profiling studies have established a critical function of wobble $U_{34}$ modification at tRNA anticodons for the efficient translation of VAA codons in yeast, nematodes and mammalian cells[17–21]. In a complementary approach, measurements of the steady-state proteome in $U_{34}$-enzyme-deficient cells have determined proteins whose abundance decreases in the absence of $U_{34}$ modification. This led to the identification of several proteins that are encoded by VAA codon-rich mRNAs and depend on $U_{34}$-enzymes for proper synthesis[19,23,27,28,36,39]. Loss of $U_{34}$-enzymes also has more pleiotropic consequences, leading to the aggregation of metastable proteins and proteotoxic stress[18–20]. However, it has been noted that high VAA codon content of mRNAs is not sufficient to predict whether the levels of the corresponding protein are decreased in $U_{34}$-enzyme-deficient cells[36]. Rather, the mild perturbation of the proteome in $U_{34}$-enzyme-deficient cells suggested that the decrease in VAA codon translation rate is too small to affect global protein synthesis[21,40]. Thus, how $U_{34}$ modification affects translational output for specific proteins and proteome subsets has remained an outstanding question.

By combining absolute quantification of proteome-wide changes in protein synthesis with steady-state proteomics and ribosome profiling, the present study provides a systematic characterization of tRNA

$U_{34}$-enzyme function in mammalian cells. Loss of $U_{34}$-enzymes leads to ribosome accumulation predominantly at lysine AAA codons. Thus, the resulting decrease in protein synthesis correlates with a high mRNA content of AAA codons and to some extent GAA codons, but not with CAA codons. This suggests that specific codons serve as better predictors for the $U_{34}$-enzyme-dependence of mRNA translation than overall VAA codon content. The synthesis of ribosomal proteins is particularly dependent on $U_{34}$-enzymes, consistent with the high AAA codon content of their mRNAs. More generally, loss of $U_{34}$-enzymes decreases the synthesis of proteins related to ribonucleoprotein complex biogenesis, RNA metabolism and sister chromatid segregation. This may reflect a requirement of these protein groups for lysine residues, whose positive charge mediates nucleic acid interactions. Thus, a substantial fraction of the proteome depends on tRNA $U_{34}$ modification for proper synthesis. However, the global decrease of protein synthesis in $U_{34}$-enzyme-deficient cells conceivably arises, in part, as a consequence of the perturbed production of ribosomes.

Although the synthesis of ribosomal proteins is substantially decreased upon loss of $U_{34}$-enzymes, their steady-state levels are only mildly affected. $U_{34}$-enzyme-deficient cells grow at a reduced rate, which might be an adaptation to largely maintain proteome homeostasis despite a decrease in protein synthesis. However, ribosomal proteins are highly abundant, constituting 4–6% of total cellular protein[3,4]. Thus, even slightly reduced production of ribosomal proteins represents a substantial decrease in biosynthetic output. The critical impact of $U_{34}$-enzymes on the proteome materializes in the context of mTORC1 inhibition. Ribosomal protein mRNAs contain TOP motifs, which renders their translation highly dependent on mTORC1 signaling[8–11]. Consequently, simultaneous suppression of mTORC1 and $U_{34}$-enzymes depletes cells of ribosomal proteins, strongly decreasing protein synthesis and inhibiting cell growth. In *S. cerevisiae*, tRNA $U_{34}$ thiolation responds to cysteine and methionine levels, thereby connecting protein synthesis to sulfur availability[24]. In *S. pombe*, reciprocal regulation of TOR signaling and Elongator regulates sexual differentiation in response to nutrient starvation[41]. Whether $U_{34}$ modification status in mammalian cells responds to nutrient supply remains unclear. Conceivably, co-regulation of mTORC1 activity and tRNA modification could be a mechanism to optimize translational output in response to changes in extracellular environment.

Yeast lacking $U_{34}$-enzymes are more sensitive to environmental stresses such as elevated temperature, various chemical stresses and nutrient deprivation[20,21,23–25,42]. This increased stress sensitivity results from perturbed translation of several VAA codon-rich stress mRNAs and from proteotoxic stress due to protein aggregation. In this context, the sensitivity of $U_{34}$-enzyme-deficient yeast to rapamycin, which induces a starvation-like state, was also interpreted as increased stress sensitivity[20,21]. However, the inactivation of mTORC1 is a protective cellular adaptation to starvation and various other stresses. By attenuating protein synthesis, mTORC1 inactivation conserves metabolic resources and prevents the accumulation of damaged or misfolded proteins[6,7,43]. mTORC1 inactivation also limits proteotoxic stress by promoting the clearance of protein aggregates through autophagy

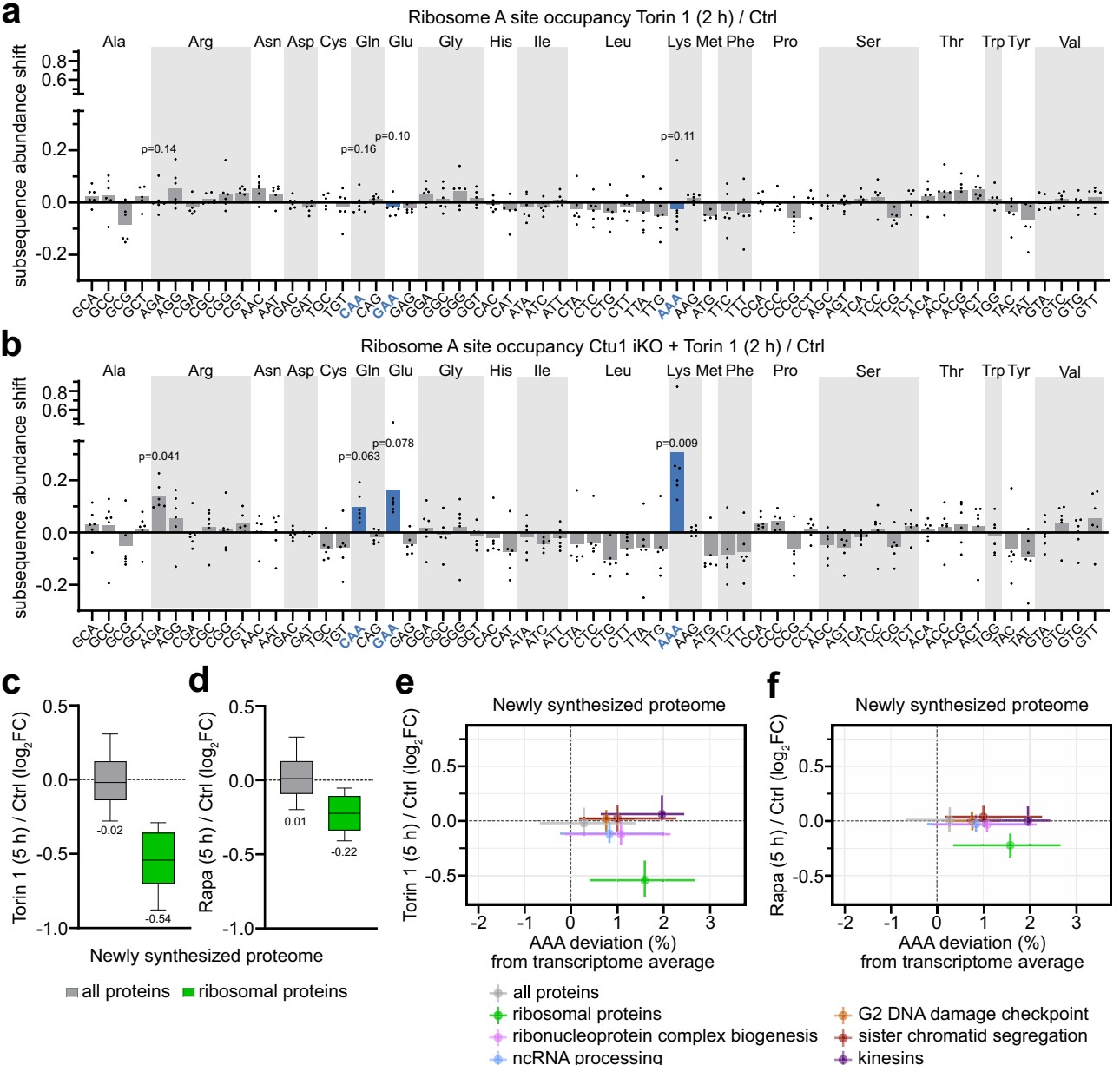

**Fig. 5 | mTORC1 and tRNA wobble enzymes converge on sustaining ribosomal protein synthesis. a, b** Codon-specific changes in ribosome A site occupancy in (**a**) control and (**b**) Ctu1 iKO EPP2 cells after 2 h ± torin 1 [300 nM], analyzed by diricore. VAA codons cognate for $U_{34}$ wobble tRNAs are highlighted. Data are represented as mean ($n = 6$ independent experiments); $p$ values were calculated by out-of-frame analysis and are shown for significant increases and for all $U_{34}$-enzyme-dependent codons. **c, d** Changes in global and ribosomal protein abundance in the newly synthesized proteome of EPP2 cells after 5 h ± (**c**) torin 1 [50 nM] and (**d**) rapamycin [50 nM], quantified by the QuaNPA workflow ($n = 3$ experimental replicates). In box plots, centre line represents median, upper and lower bounds of the box 75th and 25th percentiles, whiskers 10th to 90th percentiles. **e, f** Correlation of mRNA AAA codon usage deviation with changes in the newly synthesized proteome of EPP2 cells after 5 h ± (**e**) torin 1 [50 nM] and (**f**) rapamycin [50 nM]. Shown are protein groups whose synthesis is particularly dependent on Ctu1. Data are represented as median and interquartile range. Source data are provided as a Source Data file.

and lysosomal degradation[44]. Our work suggests an alternative explanation for the sensitivity of mammalian cells lacking $U_{34}$-enzymes to mTORC1 inhibitors. By promoting the production of ribosomal proteins, $U_{34}$-enzymes exert an anabolic role that becomes essential when mTORC1 is inactive. Thus, the potent inhibition of cell growth by concomitant suppression of mTORC1 and $U_{34}$-enzymes stems from a collapse of biosynthetic capacity.

Cancer cells dysregulate translation to promote their growth[45,46]. Many of the most commonly mutated genes in cancer are components of the growth factor signaling pathways that control protein synthesis. mTORC1 is a key driver of altered translation downstream of oncogenic signaling, in particular due to activating mutations in PI3-kinase

or loss of its antagonistic phosphatase, PTEN. By enhancing the translation of mRNAs encoding for proteins with important functions in protein synthesis and other anabolic processes, hyperactive mTORC1 promotes cancer cell growth and proliferation[10]. Increased expression of tRNA $U_{34}$-enzymes has also been observed in various cancers[22]. $U_{34}$-enzymes are required for efficient translation of several mRNAs encoding protumorigenic proteins that contribute to cancer cell survival, invasiveness and therapy resistance[19,27,28]. The present study suggests that by promoting ribosomal protein synthesis, $U_{34}$-enzymes play an important role in supporting the enhanced translation that underlies the high growth rates of cancer cells. Targeting oncogenic alterations of the protein synthesis machinery is a

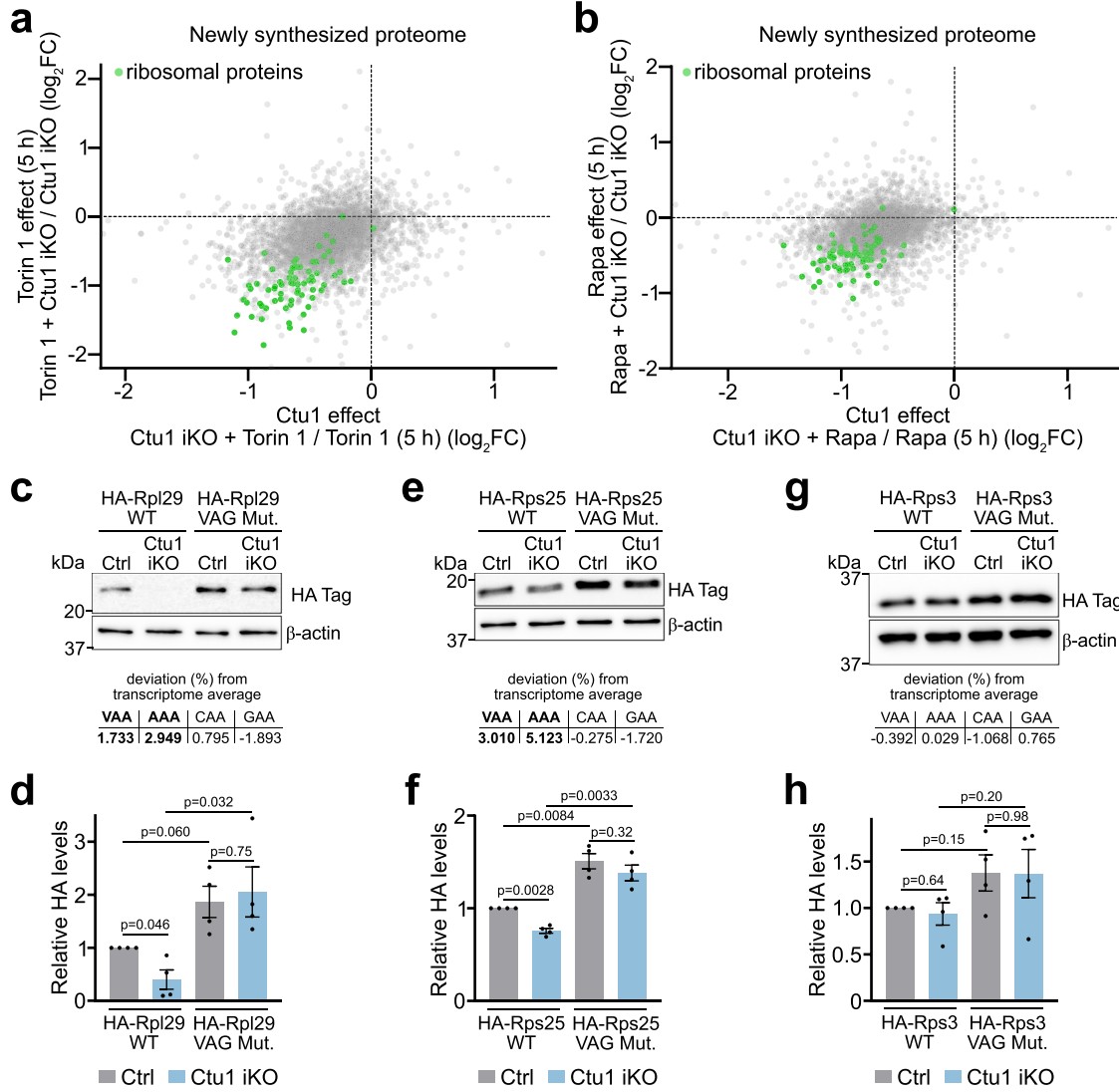

**Fig. 6 | Ctu1 promotes efficient synthesis of ribosomal proteins in a codon-dependent manner. a, b** Scatter plot comparing the newly synthesized proteome of EPP2 cells after 5 h mTORC1 inhibitor ± Ctu1 iKO (Ctu1 effect) or Ctu1 iKO ± 5 h mTORC1 inhibitor (inhibitor effect). mTORC1 inhibitors were a) torin 1 [50 nM] and (**b**) rapamycin [50 nM]. Newly synthesized proteomes were quantified using the QuaNPA workflow ($n = 3$ experimental replicates). **c–h** Expression of HA-tagged ribosomal proteins (**c**, **d**) Rpl29, (**e**, **f**) Rps25 and (**g**, **h**) Rps3 in Ctu1 iKO EPP2 cells.

Expression of wild type (WT) and VAA-to-VAG mutant (VAG Mut.) ribosomal protein variants was analyzed after 40 h + rapamycin [50 nM], MG132 [200 nM] by quantitative immunoblotting. Codon usage deviation of wild type proteins is indicated. Data are normalized to WT sgRNA controls and represented as mean ± SEM ($n = 4$ independent experiments); $p$ values were calculated by two-tailed one sample $t$-test with a hypothetical mean of 1. Source data are provided as a Source Data file.

promising approach in cancer therapy. However, although mTORC1 inhibitors potently suppress translation and growth in cultured cells, their efficacy in the treatment of human cancers has been limited[45]. The present study demonstrates that genetic ablation of U$_{34}$-enzymes renders cancer cells highly sensitive to inhibitors targeting mTORC1 or upstream growth factor signaling. Even the partial mTORC1 inhibitor rapamycin, which has only poor efficacy in many cancers[47], potently suppresses the growth of U$_{34}$-enzyme-deficient cells in vitro and in tumors. This suggests that dysregulated translational programs that promote cancer cell growth can be targeted by combined inhibition of mTORC1 and U$_{34}$-enzymes.

## Methods
### Reagents
Antibodies were from Cell Signaling (#2920 Akt (pan), #4060 Phospho-Akt (Ser473), #2217 S6 Ribosomal Protein, #2215 Phospho-S6 Ribosomal Protein (Ser240/244), #2708 p70 S6 Kinase, #9234 Phospho-p70 S6 Kinase (Thr389), #9452 4E-BP1, #5728 Elp3, #9538

Ribosomal Protein S3, #3724 HA-Tag), ProteinTech (15799-1-AP Ribosomal Protein L29), and Sigma Aldrich (A5441 β-actin, MABE343 Puromycin). The LC3 antibody was a kind gift from Tullia Lindsten. Secondary antibodies were from Sigma Aldrich (HRP anti-rabbit, HRP anti-mouse). Antibody for Flow Cytometry was from Invitrogen (17-0091-82 Cd9 Monoclonal Antibody KMC8, APC, eBioscience).

Inhibitors were from Tocris (4247 torin 1), Cayman Chemical (957054-30-7 GDC-0941, 1032350-13-2 MK-2206, 391210-10-9 PD 325901; 1211877-36-9 MG132; 88899-55-2 Bafilomycin A1), MedChem Express (HY-10219 rapamycin), Selleck Chemicals (S7101 SCH772984). Isotope-labelled amino acids were from Cambridge Isotope Laboratories (DLM-2640-1, DLM-570-PK) or Silantes (201204102, 201604102) and L-azidohomoalanine was from Jena Bioscience (CLK-AA005). E64, AEBSF, aprotinin, leupeptin, pepstatin (Serva) and benzaminidine (Sigma Aldrich) or EDTA-free protease inhibitor (Roche) were used as protease inhibitor cocktails. As phosphatase inhibitors, sodium orthovanadate, sodium fluoride, sodium pyrophosphate and sodium glycerophosphate (Sigma Aldrich) or Halt

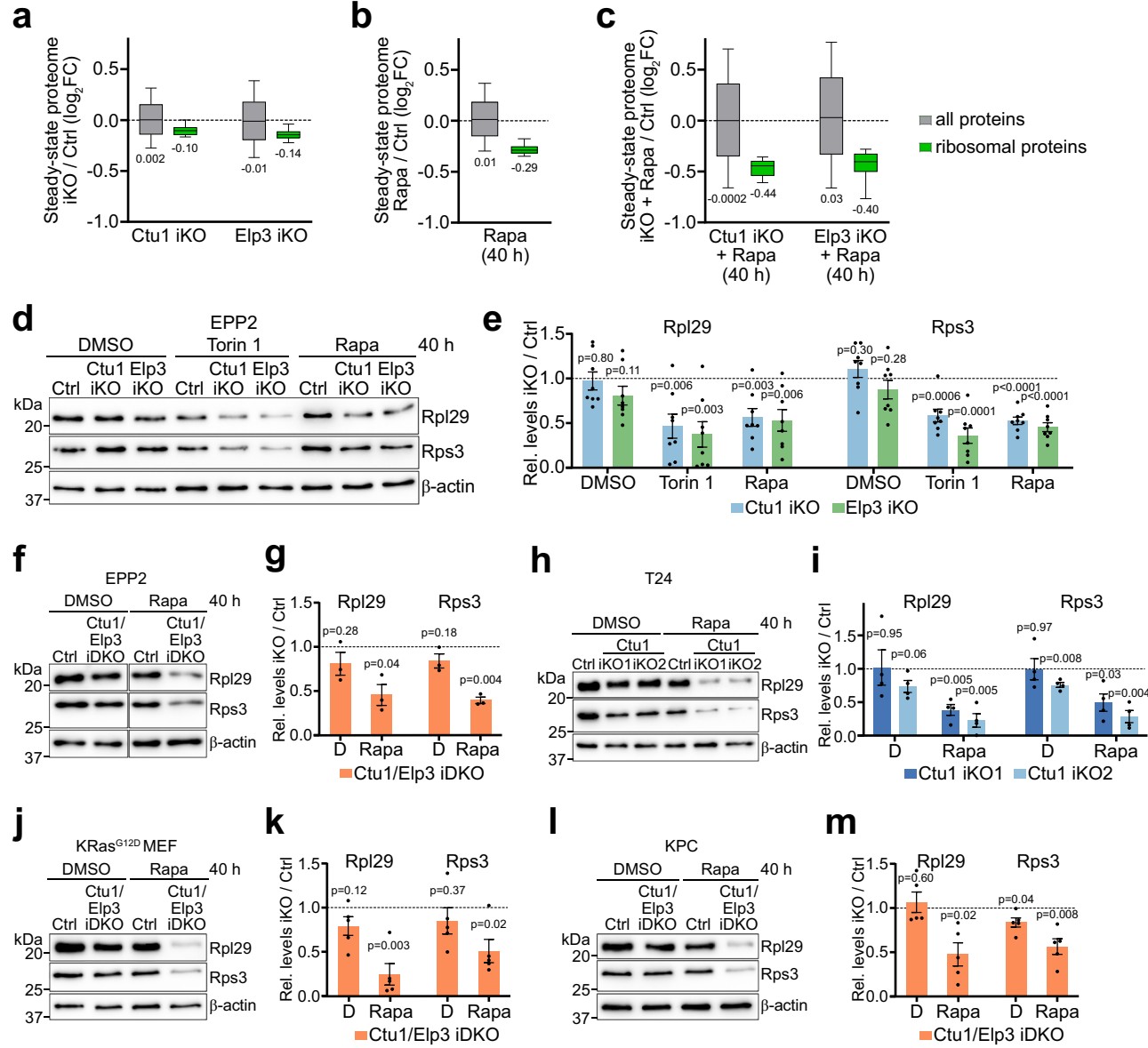

**Fig. 7 | Concerted suppression of mTORC1 and tRNA wobble enzymes depletes cells of ribosomal proteins. a–c** Changes in global and ribosomal protein abundance in the steady-state proteome of EPP2 cells. **a** Ctu1 iKO and Elp3 iKO; (**b**) control after 40 h ± rapamycin [50 nM]; (**c**) Ctu1 iKO and Elp3 iKO after ± 40 h rapamycin [50 nM]. Proteomes were quantified by label-free mass spectrometry (n = 5 independent experiments). In box plots, centre line represents median, upper and lower bounds of the box 75th and 25th percentiles, whiskers 10th to 90th percentiles. **d, e** Levels of ribosomal proteins Rpl29 or Rps3 in Ctu1 iKO and Elp3 iKO EPP2 cells after 40 h ± torin 1 [300 nM] or rapamycin [50 nM], analyzed by quantitative immunoblotting. **f–m** Changes in Rpl29 and Rps3 abundance in different $U_{34}$ enzyme-deficient cell lines after 40 h ± rapamycin [50 nM], analyzed by quantitative immunoblotting. **f, g** Ctu1/Elp3 iDKO EPP2 cells; (**h, i**) Ctu1 KO T24 cells; (**j, k**) Ctu1/Elp3 iDKO KRas^G12D MEFs; (**l, m**) Ctu1/Elp3 iDKO KPC cells. **e–m** Data are normalized to sgRNA controls (dashed line) and represented as mean ± SEM (**e** n = 8, (**g**) n = 3, (**i**) n = 4, k), **m** n = 5 independent experiments); p values were calculated by two-tailed one sample t-test with a hypothetical mean of 1. Source data are provided as a Source Data file.

Protease and Phosphatase Inhibitor Mix (Thermo Fisher Scientific) were used. Geneticin and hygromycin B were from Gibco, puromycin and blasticidin were from Santa Cruz Biotechnology, and doxycycline was from Sigma Aldrich. Most other chemicals were from Sigma Aldrich.

### Cell culture
Mouse EPP2, KRPC, SV40 large T antigen-immortalized MEFs and KPC[48], and human T24 (ATCC HTB-4) were cultured in DMEM/F-12 (Gibco 11320-074) supplemented with 10% FBS (Gibco 10270-106) and 2 mM glutamine (Gibco 25030-024). For viral production, HEK 293 T (ATCC CRL-3216) were cultured in DMEM (Gibco 41965-039)

supplemented with 10% FBS and 2 mM glutamine. Cell lines were maintained at 37 °C and 5% $CO_2$, regularly tested for mycoplasma (MycoAlert Mycoplasma Detection, Lonza) and authenticated by Single Nucleotide Polymorphism (SNP) typing by Multiplexion.

### Lentivirus production and transduction
Lentiviral vector, psPAX2 (Addgene 12260) and pCMV-VSV-G (Addgene 8454) were co-transfected into HEK 293 T cells using polyethylenimine (PEI, MW 25000, Polysciences). Viral supernatants were filtered through a 0.45 μm PES filter. Target cells were transduced at a multiplicity of infection (MOI) of 25–50% with viral supernatants containing 10 μg/ml polybrene.

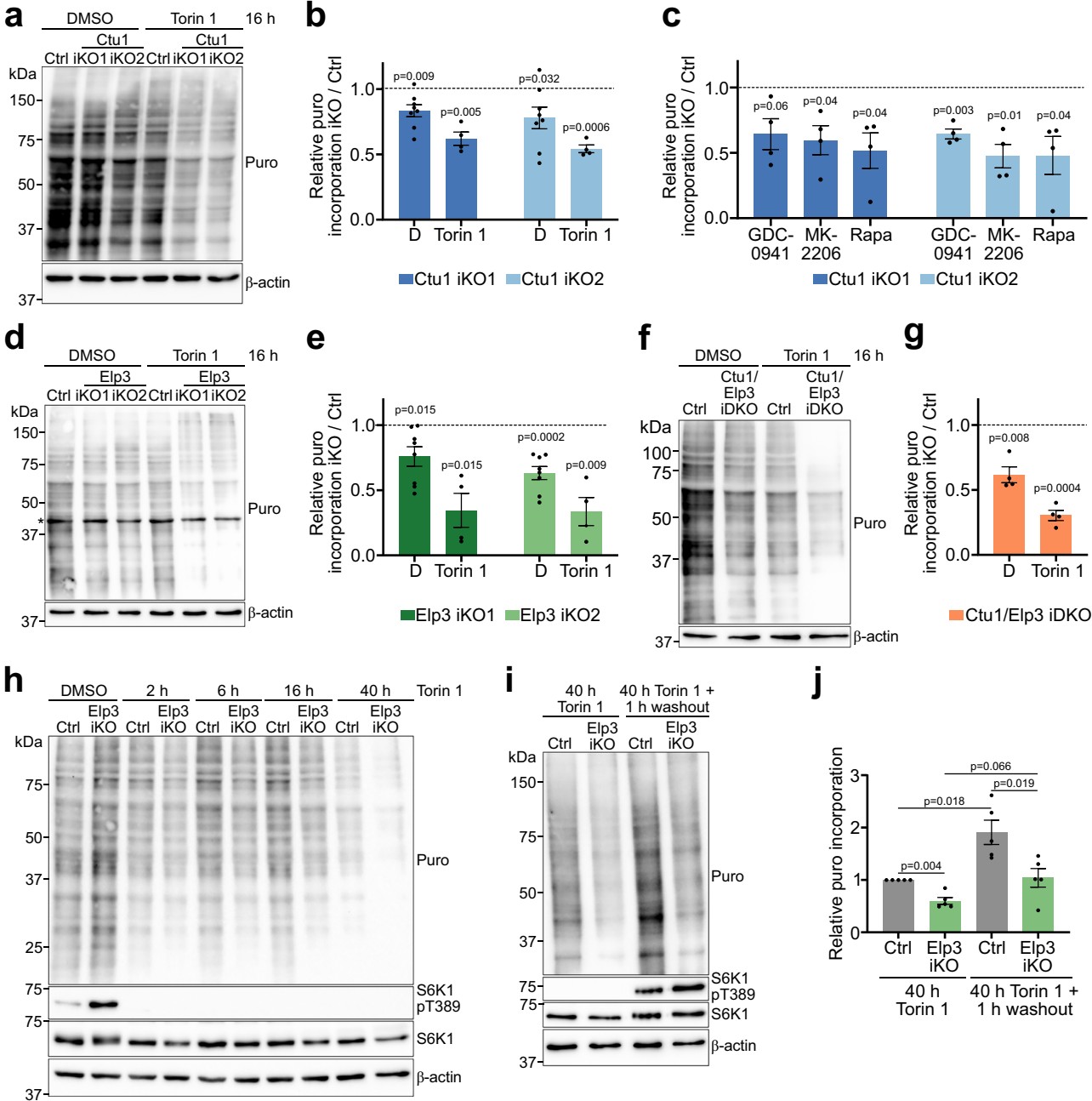

**Fig. 8 | mTORC1 cooperates with tRNA wobble enzymes to sustain translational capacity. a**–**g** Puromycin (puro) incorporation assay in $U_{34}$-enzyme-deficient EPP2 cells after 16 h inhibitor treatment, analyzed by quantitative immunoblotting. Asterisk denotes an unspecific band. **a**, **b** Ctu1 iKO ± torin 1 [300 nM]; (**c**) Ctu1 iKO ± GDC-0941 [1 μM], MK-2206 [2 μM], rapamycin [12.5 nM]; (**d**, **e**) Elp3 iKO ± torin 1 [300 nM]; (**f**, **g**) Ctu1/Elp3 iDKO ± torin 1 [300 nM]. Data are normalized to sgRNA controls (dashed line) and represented as mean ± SEM (**b**, **e**) DMSO *n* = 8, torin 1 *n* = 4, (**c**, **g**) *n* = 4 independent experiments). *p* values were calculated by two-tailed

one sample *t*-test with a hypothetical mean of 1. **h** Puromycin incorporation assay in Elp3 iKO EPP2 cells after torin 1 [300 nM] treatment for the indicated periods, analyzed by immunoblotting. **i**, **j** Puromycin incorporation assay in Elp3 iKO EPP2 cells after 40 h torin 1 [300 nM] followed by 1 h washout of torin 1, analyzed by quantitative immunoblotting. Data are normalized to sgRNA control + torin 1 and represented as mean ± SEM (*n* = 5 independent experiments); *p* values were calculated by two-tailed unpaired *t*-test with Welch correction. Source data are provided as a Source Data file.

## Generation of gene knockout cells

Doxycycline-inducible Cas9 (iCas9) cell lines were generated as previously described[29]. Briefly, EPP2, KRPC, KPC and MEFs were transduced with pRRL-SFFV-rtTA3-IRES-EcoR-PGK-HygroR, selected with hygromycin B, and transduced with pRRL-TRE3G-Cas9-P2A-BFP (EPP2, KRPC, MEF) or pRRL-TRE3G-Cas9-P2A-GFP-PGK-BlastR (KPC). To obtain single cell-derived clones, cells were sorted into 96-well plates using a FACSAria III cell sorter (BD Bioscience).

For the generation of inducible knockout (iKO), iCas9 cell lines were transduced with Dual-hU6-sgRNA-mU6-sgRNA-Ef1α-Thy1.1-P2A-NeoR or Dual-hU6-sgRNA-mU6-sgRNA-Ef1α-mCherry-P2A-PuroR harboring dual-sgRNAs (dsgRNAs) against a target gene. dsgRNA-positive cells were selected with geneticin or puromycin. For inducible double knockout (iDKO), two dsgRNA plasmids were sequentially transduced and selected with antibiotics. Cas9 expression was induced by treating the cells with 300 ng/ ml doxycycline for 3 and 4 days. For the

generation of constitutive knockouts, cells were sequentially transduced pLenti-Cas9-BlastR (Addgene 52962) and Dual-hU6-sgRNA-mU6-sgRNA-Ef1a-Thy1.1-P2A-NeoR harboring dsgRNAs against the target gene.

Clonal Ctu1 knockout cells were generated by co-transfection of iCas9 EPP2 or iCas9 KRPC cells with a mixture of pSpCas9(BB)-2A-GFP (PX458, Addgene 48138) harboring either of the Ctu1 dsgRNA2 sequences or Rosa26 sgRNA sequences, as control (Suppl. Data 8). GFP-positive cells were sorted into 96-well plates using a FACSAria III cell sorter (BD Bioscience) to obtain single clones. Successful editing of the *Ctu1* was confirmed by genomic PCR.

Controls for CRISPR/Cas9 experiments were dsgRNAs targeting non-coding chromosome regions (Chr1.1, Chr1.3 or Rosa26) or cells without doxycycline treatment. sgRNA sequences are listed in Suppl. Data 8.

### Generation of Ctu1 and Elp3 overexpression constructs

A mouse Ctu1 ORF was obtained from Horizon Discovery; a human Elp3 ORF was obtained from the genomics core facility at the DKFZ. Ctu1 and Elp3 were cloned into pLV-Ef1a-C-Flag-IRES-NeoR (derived from Addgene 85139). Primers for ORF amplification were designed using the NEBuilder Assembly Tool, cloning was performed using the HiFi DNA Assembly Master Mix (NEB).

### Generation of tagged ribosomal protein constructs

For the generation of N-terminal HA-tagged ribosomal proteins, Rpl29 and Rps3 were amplified from an EPP2 cDNA library. Rps25 and all VAG mutant variants, in which mRNA VAA codons were replaced by synonymous VAG codons, were synthesized by Genewiz. The ribosomal protein variants were cloned into pLV-EF1a-N-HA-IRES-NeoR (derived from Addgene 85139) or pLV-UbC-IRES-BlastR (derived from Addgene 85113), followed by replacement of the EF1a or UbC promoter sequences by the respective endogenous 5′ UTR sequence (+ 500 bp upstream for Rpl29 and Rps3) containing a TOP motif. Primers for cDNA amplification and introduction of the N-HA tag for each ribosomal variant as well as for amplification of 5′ UTR sequences from genomic DNA of EPP2 cells were designed using the NEBuilder Assembly Tool, cloning was performed using the HiFi DNA Assembly Master Mix (NEB).

### PCR of genomic DNA

Genomic DNA was isolated either using the DNeasyBlood & Tissue Kit (Qiagen 69504) according to manufacturer instructions or as described in the following. Cells were resuspended in DNA lysis buffer (0.45% Tween 20, 0.45% Triton X-100, 2.5 mM $MgCl_2$, 50 mM KCl, 10 mM Tris-HCl pH 8, 100 ug/mL Proteinase K). The solution was incubated for 1 h at 56 °C, followed by 10 min at 95 °C and stored at −20 °C. Successful editing was confirmed by PCR using genotyping primers that amplify the genomic *Ctu1* locus containing the cutting sites of the two sgRNA sequences (Ctu1_FWD AGACTCCCCAGATGACTCCC, Ctu1_REV CAGCCCACCCCACTATCTTG). PCR was performed using Taq DNA polymerase (Thermo Fisher Scientific 10342020) following standard protocols.

### Proliferation-based CRISPR-Cas9 screen

For the CRISPR-Cas9 screen, the Vienna genome-wide sgRNA library was packaged into lentiviral particles, and EPP2 iCas9 cells were transduced at a MOI of less than 0.15 with 500- to 1000-fold library representation. Selection of sgRNA-positive cells was started after one day with 200 μg/mL geneticin. Successful selection was confirmed by immunostaining and flow cytometry detection of Thy1 expression. Cas9 expression in selected EPP2 cells (>94% Thy1-positive) was induced by 300 ng/ml doxycycline two days before the onset of the screen. At Day 0, cells were plated in medium containing 300 nM torin 1 or DMSO; doxycycline was added for two more days. The CRISPR-Cas9 screen was performed with amino acid-free, glucose-free DMEM/

F-12 (US Biological D9807-11), pH 7.3, which was supplemented with glucose and amino acids at standard DMEM/F-12 concentrations, 5% FBS and physiological albumin levels (2%; Sigma Aldrich A1470). Cells were cultured at a library representation of at least 600-fold. DMSO-treated cells were replated every second day with an additional media change after 1 day. Torin 1-treated cells were replated every 3 days with an additional media change after 1.5 days. The starting cell population was harvested in two replicates for sequencing (T0), final cell populations were harvested after 14 population doublings. Cells were washed with PBS, pelleted and stored at −80 °C until DNA extraction. At the time of harvest and for torin 1 at an intermediate time point, cells were additionally plated to harvest protein samples for western blot analysis.

### Production of NGS libraries

Next-generation sequencing (NGS) libraries of CRISPR-Cas9 screen samples were prepared as described previously[48,49]. Briefly, genomic DNA was isolated by lysing cells in 10 mM Tris-HCl, 150 mM NaCl, 10 mM EDTA, 0.1% SDS. This was followed by proteinase K treatment, DNAse-free RNAse digest, phenol extraction and DNA precipitation with 2-propanol (performed twice). After several freeze-thaw cycles, sgRNA cassettes were amplified by nested PCR. For each sample, barcoded NGS libraries were generated using a two-step PCR protocol. In the first PCR, 0.2 μl of AmpliTaq Gold (Thermo Fisher Scientific 4311820) was used in 50 μl reactions containing 1 μg of genomic DNA. The PCR product was pooled for each sample and purified using AMPure XP magnetic PCR purification beads (Beckman Coulter A63881). In the second PCR, purified products (10 ng DNA) were used to incorporate standard Illumina adapters. The final libraries were pooled and sequenced on a NovaSeq6000 platform (Illumina) according to the manufacturer's protocol. Primers used for NGS library preparation were: PCR1 FWD1 GCATACGAGATAGCTAGCCACC, REV1 CTCTTTCCCTACACGACGCTCTTCCGATCTNNNNNNXXXXTTCCAGC ATAGCTCTTAAAC (NNNNNN: random nucleotides, XXXX: sample specific barcode); PCR2 FWD2 CAAGCAGAAGACGGCATACGAGA-TAGCTAGCCACC, REV2 AATGATACGGCGACCACCGAGATCTACACTC TTTCCCTACACGACGCT.

### Analysis of pooled CRISPR-Cas9 screens

For the quantification of raw sequencing reads, the crispr-process-nf Nextflow workflow available at https://github.com/ZuberLab/crispr-process-nf was used. As described previously[48,49], all guides in the library were padded with Cs to equal length before creating an index for Bowtie 2 (v2.3.0). Random 6mer nucleotides were trimmed using the fastx_trimmer from the fastx-toolkit (v0.0.14) (http://hannonlab.cshl.edu/fastx_toolkit/) before demultiplexing by 4mer sample barcodes with fastx_barcode_splitter (−mismatches 1 −bol). Afterwards, barcodes and 20mer spacers were trimmed using fastx_trimmer. Reads were aligned with Bowtie 2 (-L 18 −score-min 'C,0, − 1' -N 0 −seed 42) and quantified with featureCounts (v1.6.1)[50].

Gene-level enrichment or depletion of sgRNAs was calculated using MAGeCK (v0.5.9)[30] and the crispr-mageck-nf Nextflow workflow, available at https://github.com/ZuberLab/crispr-mageck-nf. First, count tables were filtered to remove sgRNAs with less than 50 counts in samples at T0. Read counts were median normalized and average $log_2$ fold changes, *p* values and false discovery rates (FDRs) were calculated. To determine enrichment or depletion of sgRNAs for each condition and time point, sgRNAs abundance was either compared to T0, or between the endpoint conditions (torin 1 versus DMSO).

### Cell proliferation assay

For proliferation assays, cells with the different genotypes and inhibitor treatments were counted using a CASY Cell Counter and Analyzer (OMNI Life Sciences). Cas9 expression was induced by treating cells with 300 ng/ml doxycycline for 3 and 4 days prior to the onset of the

proliferation assay. Data are shown as fold change in cell number from day 0 to day 3. Screen hits were validated using competitive proliferation assays. iCas9 cells carrying either a dsgRNA-mCherry construct or no sgRNA (control) were treated with doxycycline (300 ng/ml) for 2 days. At the onset of the competitive proliferation assay, the two cell populations were mixed and co-cultured in 300 nM torin 1 or DMSO, 2% albumin. Doxycycline was also added for the first 2 days. The percentage of dsgRNA-mCherry-positive cells was assessed by a Guava easyCyte flow cytometer (Merck) on day 0 and at each passage.

### Animal experiments
For orthotopic tumor experiments, Ctu1 iKO and control EPP2 cells were injected into the pancreas of mice. Surgeries were performed under isoflurane anesthesia on a heated plate. In vivo experiments were performed with 16 and 24 week-old (mixed between groups) female and male (mixed between groups) C57BL/6J Rag2$^{-/-}$ mice. Animals were bred in-house (Research Institute of Molecular Pathology (IMP), Vienna Bio-Center (VBC), Vienna, Austria) and housed under standard pathogen-free conditions at a temperature of $22 \pm 1\,°C$, $55 \pm 5\%$ humidity and a photoperiod of 14 h light: 10 h dark with food and water given *ad libitum*. Animal experiments were performed in accordance with a protocol approved by the local ministry (Research Institute of Molecular Pathology (IMP), Vienna BioCenter (VBC), Vienna, Austria). A small incision on the upper left quadrant of the shaved abdomen was made and the spleen identified, followed by externalization of the pancreas. $5 \times 10^5$ cells suspended in 10 µl PBS were injected into the pancreas using a Hamilton syringe. Organs were carefully re-situated, and the peritoneum closed with a resorbable 6-0 Vicryl suture, followed by skin closure with sterile wound clips. Animals received intraperitoneal injections with 5 mg/kg carprofen preemptively and every 12 h for 48 h after surgery. Health status of mice was monitored daily. Starting on day 4 post injections, the mice were randomly divided into treatment and vehicle cohorts. The treatment cohort received 5 mg/kg/day rapamycin (MedChemExpress HY-10219) in 50 µl i.p. for 9 days, the vehicle cohort received 50 µl vehicle (10% DMSO, 40% PEG300, 5% Tween-80 in PBS). Mice were sacrificed 14 days after orthotopic injection, and the tumor weight was measured.

### Immunoblotting
Cells were washed with ice-cold PBS and subsequently lysed using ice-cold lysis buffer (50 mM HEPES pH 7.4, 40 mM NaCl, 2 mM EDTA, 1 mM Na orthovanadate, 50 mM NaF, 10 mM Na pyrophosphate, 10 mM Na glycerophosphate, 1% Triton X-100, 1× Halt protease and phosphatase inhibitor cocktail) for 15 min. Soluble lysate fractions were prepared by centrifugation at $18,000 \times g$ for 5 min. Protein concentrations were determined using the Pierce BCA Protein Assay (Thermo Fisher Scientific). Equal amounts of protein were subjected to analysis through SDS gel electrophoresis and immunoblotting on nitrocellulose membranes following standard protocols. When multiple antibodies were used on separate membranes, sample preparation, electrophoresis and immunoblotting were carried out in parallel under identical conditions. To remove antibodies, membranes were incubated for 20 min in stripping buffer (0.2 M glycine, 0.1% SDS, pH 2.2), washed three times for 10 min with TBST (20 mM Tris, 150 mM NaCl, 0.1% Tween-20, pH 7.5), and re-blocked. Blots were imaged with ECL substrates Clarity Max Western (BioRad) on a ChemiDoc Touch imaging system (BioRad). Western blots were quantified using Image Lab software (BioRad). For each sample, band intensities were determined using the adjusted total lane volume intensity normalized to the corresponding β-actin intensity. Uncropped blots are provided in the Source Data file.

### Puromycin incorporation assay
For puromycin incorporation assays, cells were treated with puromycin [83 µM] for 10 min. Preparation of cell lysates and

immunoblotting was performed as described above. Puromycin incorporation was detected with anti-puromycin antibody. In parallel, western blotting of β-actin was performed under identical conditions and used as normalization control. For quantification, the total lane band intensity was quantified as above. Note that incubation with secondary antibody gave rise to an unspecific band at ~40 kDa, which was excluded from quantifications.

### mTORC1 re-activation assay
Cells were treated with torin 1 for 40 h. To acutely re-activate mTORC1, torin 1 was washed out by rinsing cells once with media containing 5 mg/mL fatty acid-poor BSA (Sigma Aldrich 126579), followed by 1 h incubation in fresh BSA-supplemented medium. To reduce BSA background, cells were washed three times with ice-cold PBS prior to lysis.

### Ribosome profiling library preparation
Cells were treated for 2 h with 300 nM torin 1 or DMSO. Experiments were performed in two biologically independent replicates on consecutive days. Samples were processed for ribosome profiling as previously described[35]. In brief, cells were washed with ice-cold PBS and 100 ug/ml cycloheximide. After scraping and pelleting, cells were lysed in 20 mM Tris-HCl pH 7.5, 10 mM MgCl$_2$, 100 mM KCl, 1% Triton-X 100, 2 mM DTT, 100 ug/ml CHX and 1× protease inhibitor cocktail. After 10 min on ice, lysates were centrifuged at 6400 rpm for 5 min. The supernatant was digested with 2 U/µl RNase I (Ambion AM2294) for 45 min at room temperature. For RNA-seq, 10 µl of the supernatant was added to 1 mL Trizol (Thermo Fisher Scientific) and stored at −80 °C until further processing. For ribosome profiling, lysates were separated on a linear sucrose gradient ranging from 7% to 47% using a SW-41 Ti rotor (Beckman) for 2 h at 36,000 rpm. Fractions that contained high concentrations of monosomes were combined and treated with proteinase K, 1% SDS for 1 h at 45 °C. The released RNA fragments were purified using Trizol reagent and precipitated with the addition of glycogen.

The ribosome profiling library preparation was performed as previously described[35,51]. In brief, ribosome protected fragments (RPFs) were size-selected in range of 17 to 34 nt. Resulting fragments were dephosphorylated at the 3′end using T4 PNK (New England Biolabs M0201S) for 1 h at 37 °C. Pre-5′adenylated adaptors were ligated to the RPF 3′ends using T4 RNA Ligase 2 truncated KQ (New England Biolabs M0351L) for 3 h at 22 °C. Subsequently, samples were treated with 5′ yeast deadenylase and RecJ exonuxlease (each 10 U/µl, New England Biolabs M0331S, M0264S) for 45 min at 30 °C to eliminate unbound adaptors. Next, rRNA homologous, biotinylated RNA oligos were added to the samples and incubated for 1 min at 95 °C, followed by 15 min at 37 °C. Subsequently, MyOne Streptavidin magnetic beads (Invitrogen 65001) were added and incubated for 30 min at 37 °C. Following magnetic separation, the RNA was precipitated and subjected to reverse transcription using the SuperScript III First-Strand Synthesis Kit (Invitrogen 18080051). Samples were incubated for 50 min at 50 °C and then treated with 5 M NaOH for 3 min at 95 °C to eliminate RNA. Remaining cDNA was circularized with the CircLigase II Kit (LGC BioSearch CL9021K). The circular cDNA was subjected to PCR to introduce Illumina i7 indices using the Q5 High Fidelity 2× MasterMix (NEB M0492S) and was size-selected with an 8% non-denaturing PAGE gel. The extracted and precipitated DNA was resuspended in nuclease-free water and quantified with the Qubit (Invitrogen). DNA sample pools were adjusted to a library concentration of 2 nM and subjected to deep-sequencing on a NextSeq2000 platform (Illumina) at the DKFZ Sequencing Open Lab, associated with the DKFZ Genomics & Proteomics Core Facility.

### Ribosome profiling data analysis
The FASTQ raw data was provided by the DKFZ Genomics & Proteomics Core Facility and analyzed as follows: Adapters were trimmed

using cutadapt (v3.4), followed by demultiplexing with barcode_splitter from FASTX-toolkit (v0.0.6). Fragments smaller than 27 nt were discarded. Unique Molecular Identifiers were extracted using umi_tools (v1.1.1). Subsequently, rRNA reads were detected and dropped from the processing using BLAST-Like Alignment Tool (BLAT) (v36x2). The rRNA index for RNA18S5, RNA28S5 and RNA5-8S5 was constructed manually from NCBI RefSeq annotation. The filtered reads were aligned using Spliced Transcripts Alignment to a Reference (STAR) (v2.5.3a) to GRCh37/hg19 with '–quantMode TranscriptomeSAM GeneCounts –outSAMmapqUnique 0'.

Differential ribosome codon reading (diricore) analysis[35] using the diricore pipeline available at https://github.com/A-X-Smitt/B250_diricore was performed to determine ribosome occupancy in amino acid codons from filtered RPF reads. Using gene-level normalization, codon frequency biases were ruled out. Reads based on unambiguous gene IDs were quantified for reading frame counts in position 12 and 15 downstream of the 5′ end, representing ribosomal A- and P-site. Note that the ribosome P-site did not show any obvious changes upon loss of $U_{34}$-enzymes or mTOR inhibition. Subsequences analysis of ribosomal transcripts was performed by extracting read information aligning to the respective transcripts from BAM files. Resulting subset BAM files were used as new subsequence analysis input.

## Out-of-frame analysis for diricore

To evaluate the significance of diricore signals, we generated a background distribution of values for each dataset by shifting nucleotides (−1/+1) from the analyzed codon. This distribution was then used to perform Z-tests of the diricore signal at the original position (+0). We confirmed that the background distribution followed an approximately normal distribution by visually inspecting q–q plots and performing the Anderson–Darling and Shapiro–Wilk tests.

## RNA isolation and RNA sequencing

Cell lysates supplemented with Trizol (from ribosome profiling experiments) were treated with chloroform. After centrifugation for 15 min at 14,000 rpm, RNA was precipitated with isopropanol. The pellet was washed twice with 70% ethanol. The RNA pellet was air-dried and resuspended in RNase-free water. RNA yield was quantified using a Qubit (Invitrogen). Library preparation and RNA sequencing was performed at the NGS Core Facility, German Cancer Research Center (DKFZ). Sequencing libraries were prepared using the TruSeq mRNA stranded Kit (Illumina) according to manufacturer instructions. In brief, mRNA was purified from 500 ng of total RNA using oligo(dT) beads. Then, poly(A) + RNA was fragmented to 150 bp and converted to cDNA. The cDNA fragments were then end-repaired, adenylated on the 3′ end, adapter ligated and amplified with 15 cycles of PCR. The final libraries were validated using Qubit (Invitrogen) and Tapestation (Agilent Technologies). 2 × 100 bp paired-end sequencing was performed on a NovaSeq 6000 platform (Illumina) according to the manufacturer's protocol.

## RNA sequencing data analysis

The FASTQ raw data was provided by the DKFZ Genomics & Proteomics Core Facility as described (https://github.com/DKFZ-ODCF/RNAseqWorkflow). Differential gene expression analysis of the RNA sequencing results was performed using raw read counts (≥10 per gene) using DESeq2 (version 1.30.1) with default settings.

## Sample preparation for steady-state proteomics

For steady-state proteomics, cells were treated with rapamycin (50 nM) or DMSO for 40 h, with media change after 24 h. For protein extraction, cells were washed twice with ice-cold PBS and lysed using RIPA buffer (150 mM NaCl, 50 mM Tris-HCl pH 8, 1% NP-40, 0.5% sodium deoxycholate, 0.1% SDS, 1 mM EDTA, 5 mM $MgCl_2$, 1× protease inhibitor cocktail). E64, AEBSF, aprotinin, leupeptin, pepstatin (Serva)

and benzamidine (Sigma Aldrich) were used as protease inhibitor cocktail. After adding Benzonase (250 U/mL), lysates were vortexed and incubated on ice for 15 min. Soluble lysate fractions were prepared by centrifugation at 18,000 × g for 15 min. 5 independent biological replicates were collected on consecutive days. Protein concentrations were determined with the Pierce BCA Protein Assay (Thermo Fisher Scientific). Equivalent protein amounts were subjected to MS in a block randomized manner.

## LC-MS/MS analysis for steady-state proteomics

10 μg proteins were digested with trypsin using a AssayMAP Bravo liquid handling system (Agilent technologies) running the autoSP3 protocol[52]. A 60 min LC-MS/MS analysis was carried out on an Ultimate 3000 UPLC system (Thermo Fisher Scientific) connected to an Orbitrap Exploris 480 mass spectrometer. Peptides were online desalted on a trapping cartridge (Acclaim PepMap300 C18, 5 μm, 300 Å wide pore; Thermo Fisher Scientific) for 3 min using 30 μl/min flow of 0.1% (v/v) trifluoroacetic acid in water. The analytical multistep gradient (300 nl/min) was carried out on a nanoEase MZ Peptide analytical column (300 Å, 1.7 μm, 75 μm × 200 mm; Waters) using solvent A (0.1% (v/v) formic acid in water) and solvent B (0.1% (v/v) formic acid in acetonitrile). After 4 min at 2% B, for the analytical separation the concentration of B was linearly ramped from 5% to 30% over 43 min. The end of the analysis included washing (78% B for 2 min) and equilibration (2% B for 10 min). Eluting peptides were analyzed in the mass spectrometer using data independent acquisition (DIA) mode. A full scan at 120k resolution (350–1400 m/z, 300% AGC target, 45 ms maxIT. profile mode) was followed by 20 windows of variable isolation width (400–1000 m/z, 1000% AGC target, 30k resolution, 54 ms maxIT, centroid, 1 Da overlap) for fragment spectra acquisition. Collision energy was set at 28%. After each sample, wash runs were performed to avoid sample carryover and regular QC runs ensured high and consistent MS performance.

## Steady-state proteomics data analysis

Analysis of DIA RAW files was performed with Spectronaut (Biognosys, version 17.1.221229.55965) in directDIA+ (deep) library-free mode. Default settings were applied with the following adaptions. Within the pulsar search in result filters the m/z max was set to 1800 and min to 300, relative intensity was set to 5%. Within DIA analysis under quantification, the protein LFQ method was set to MaxLFQ. The data were searched against the mouse proteome from Uniprot (mouse reference database with one protein sequence per gene, containing 21,957 unique entries from 03.05.2023) and the contaminants FASTA from MaxQuant (246 unique entries from 22.12.2022).

## Sample preparation for nascent proteomics

For nascent proteomics, cells were treated with SILAC medium (DMEM-F12 without methionine, lysine and arginine, 5% dialyzed FBS and 200 mg/l proline) in the presence or absence of 50 nM torin 1, 50 nM rapamycin or DMSO for 45 min. Afterwards, cells were labeled in SILAC medium supplemented with intermediate stable isotope-containing lysine and arginine ($^{13}C_6$-Arg, Lys-D4) or heavy stable isotope-containing lysine and arginine ($^{13}C_6$, $^{15}$N4-Arg, Lys-D9), L-azidohomoalanine (100 μM AHA) and the different mTORC1 inhibitors. For relative SILAC quantification, the two conditions that were compared were labeled with medium or heavy isotopes, respectively.

Sample preparation and enrichment was performed using the QuaNPA workflow[34]. Samples were collected in 3 experimental replicates. For protein extraction, cells were washed twice with ice-cold PBS and lysed in 300 mM HEPES (pH 8.0), 1% SDS and 1× protease inhibitor cocktail. E64, AEBSF, aprotinin, leupeptin, pepstatin (Serva) and benzamidine (Sigma Aldrich) were used as protease inhibitor cocktail. Cell lysates were sonicated in AFA-tube TPX strips (Covaris), using a Covaris LE220R-Plus. Protein concentrations of the sonicated lysates

were determined with the Pierce BCA Protein Assay (Thermo Fisher Scientific). A total of 200 µg (100 µg per condition) protein was used as input for the enrichments. The combined lysates of the two conditions that were compared (medium and heavy isotopes) were mixed and diluted to a total volume of 150 µl using lysis buffer. For the automated enrichment of newly synthesized proteins, a Bravo liquid handling system (Agilent technologies) was used. Samples were partially alkylated by adding 3.4 µl of 600 mM iodoacetamide (IAA) for 20 min at room temperature. Subsequently, 20 µl of magnetic alkyne agarose (MAA) beads in a 20% (v/v) suspension, diluted in lysis buffer, and the Copper(I)-catalyzed Azide Alkyne Cycloaddition (CuAAC) reaction mixture (21.62 mM $CuSO_4$, 108.11 mM Tris-hydroxypropyltriazolylmethylamine (THPTA), 216.22 mM pimagedine hydrochloride, 216.22 mM sodium ascorbate) were added and incubated for 2 h at 40 °C. After coupling of AHA-containing newly synthesized proteins, beads were washed with 150 µl $H_2O$. Bound proteins were reduced and alkylated by adding 150 µl of 10 mM Tris(2-carboxyethyl)phosphine (TCEP) and 40 mM 2-chloroacetaminde (CAA), dissolved in 100 mM Tris-HCl (pH 8.0), containing 200 mM NaCl, 0.8 mM EDTA, 0.8% SDS and incubated for 20 min at 70 °C, followed by 15 min at 20 °C. Beads were subsequently washed, and resuspended in 50 µl 100 mM ammonium bicarbonate (pH 8.0). Proteins were digested off the beads with 0.5 µg trypsin and peptides subsequently purified running a modified version of the autoSP3 protocol[52]. Peptide-containing supernatants were frozen at −80 °C and subsequently lyophilized using a UNIVAPO-150H vacuum concentrator, coupled to a UNICRYO MC2 cooling trap and UNITHERM 4/14 D closed circuit cooler (UNIEQUIP). Magnetic carboxylate Sera-Mag Speed Beads (Fischer Scientific) were diluted to 100 µg/µl in 10% formic acid and 5 µl were added to each lyophilized sample. Aggregation of the peptides onto beads was induced via addition of 195 µl acetonitrile and incubating for 18 min, while shaking at 100 rpm on the orbital shaker. Next, the supernatant was removed, the beads were washed twice with 180 µl acetonitrile and subsequently dried. Finally, the beads were resuspended in 20 µl 0.1% formic acid in water and sonicated in an Ultrasonic Cleaner USC-T (VWR) for 10 min. The purified peptides were dissolved in 0.1% formic acid and used for LC-MS/MS analysis.

For normalization of the newly synthesized proteome, non-enriched pulsed SILAC and AHA-labelled proteins cell lysates were processed. 50 µg of isotope-AHA-labelled cell lysate was collected by pooling equal parts of the cell lysates of the two conditions that were compared. The manual SP3 protein protocol[53] was used for the preparation of proteomics samples. Aggregation of proteins on the magnetic beads was induced by addition of acetonitrile to a final concentration of 50% (v/v) and incubating at room temperature for 18 min. The beads were subsequently washed twice with 80% (v/v) ethanol and acetonitrile. On bead digestion of proteins (trypsin) in the samples was carried out in 100 mM ammonium bicarbonate (pH 8.0) for 16 h at 37 °C. Following the tryptic digestion, the supernatant was removed from the beads using a magnetic rack and trifluoroacetic acid (TFA) was added to a final concentration of 1% (v/v).

## LC-MS/MS analysis for nascent proteomics

LC-MS/MS was carried out on an EASY-nLC 1200 system (Thermo Fischer Scientific) connected to a QExactive HF mass spectrometer (Thermo Fischer Scientific). Peptides were online desalted on a trapping column (Acclaim PepMap C18, 20 mm × 100 µm, 5 µm C18 particles, 100 Å wide pore, Thermo Fischer Scientific) with constant flow of solvent A (0.1% formic acid) at a maximum pressure of 800 bar. The analytical multistep gradient (300 nL/min) was carried out on a nanoEase m/z peptide BEH C18 analytical column (250 mm × 75 µm 1/PK, 130 Å, 1.7 µm, Waters) using solvent A (0.1% formic acid) and solvent B (80% acetonitrile) as mobile phases. The analytical column was equilibrated with 2 µL A at a maximum pressure of 600 bar and heated to 55 °C using a HotSleeve+ column oven (Analytical Sales & Services). Concentration of B was gradually

increased during the elution of the peptides. The gradient used for the analysis of the newly synthesized proteome and input samples started with 4% B and was increased to 6% in the first 1 min, increased to 27% at 51 min and further increased to 44% at 70 min. After 70 min the percentage of B was raised to 95%. After 80 min the system was re-equilibrated (5% B for 10 min). After each sample, wash runs were performed to avoid sample carryover and regular QC runs ensured high and consistent MS performance.

Eluting peptides were analyzed in the mass spectrometer using data independent (DIA) mode. The full scan at 60000 FWHM resolution (400–1000 m/z, 3e6 AGC target, 20 ms maxIT) was followed by 26 windows of equal size (30000 FWHM, 1e6 AGC target, 40 ms maxIT, 1 Th overlap) with a width of 23.3 m/z, covering the scan range from 402 to 982.8 m/z. Collision energy was set to 27 and a fixed first mass of 200 m/z was set for the acquisition of the MSMS spectra.

## Nascent proteomics data analysis

Raw files from newly synthesized proteome measurements were analysed using DIA-NN (version 1.8.1)[54]. A murine proteome FASTA file, retrieved from the SwissProt database (version from 12.02.21 with 17,056 entries) was used for processing of the raw data. Additionally, a FASTA file containing common protein contaminants was added for the spectral library prediction. A predicted spectral library was generated from the FASTA files. Default settings were used for the spectral library prediction, with the addition of methionine oxidation as variable modification. For the processing of the raw files, the default settings of DIA-NN were used with additional functions from the plexDIA module enabled. Three SILAC channels with mass shifts corresponding to Lys, Lys4 (+4.025107 Da), Lys9 (+9.056356 Da), Arg, Arg6 (+6.020129 Da), Arg10 (+10.008269 Da) and an additional decoy channel with lysine (+12.0033 Da) and arginine (+13.9964 Da) were registered. Translation of retention times between peptides within the same elution group was enabled. The first $^{13}C$-isotopic peak and monoisotopic peak was included for the quantification and the MS1 deconvolution level was set to 2. Peptide length range was set from 7 to 30, precursor charge rate was set from 1 to 4, mass of charge (m/z) range of the precursors was set from 300 to 1800 and fragment ion m/z range was set from 200 to 1800. Precursor FDR was set to 1%. Precursor matrix output tables were filtered for FDR < 0.01 and additionally for channel $q$-value < 0.01 and translated $q$-value < 0.01. The MBR function in DIA-NN was enabled. The "report.pr_matrix_channels_translated.tsv" and "report.pr_matrix_channels_ms1_translated.tsv" output tables were processed in the R software environment (version 4.0.3) using custom scripts. Identified contaminants were removed and protein abundance was calculated using the MaxLFQ algorithm, applied to the individual SILAC channels, using the iq (version 1.9.6) R package function "process_long_format()". For MS1- and MS2-based quantification, the "Ms1.translated" and "precursor.translated" quantity was used for the MaxLFQ calculation, respectively. Protein-group SILAC ratios were calculated for each sample using the LFQ values. For differential expression analysis, only unique protein groups (single Uniprot identifier) with a minimum of 2 SILAC ratios values in 3 replicates were used. Average SILAC ratios for commonly identified newly synthesized proteins were determined in the pooled non-enriched samples, via MS1-based quantification and were used to determine the average rate of protein synthesis in the respective condition. This average ratio between the respective conditions was used to center the distribution of SILAC ratios in the newly synthesized proteome data.

## Statistical analysis of proteomics

Differential expression tests were carried out using Limma (version 3.46.0)[55] and DEqMS (version 1.8.0)[56] R/Bioconductor packages, by fitting the data onto a linear model and performing an empirical Bayes moderated $t$-test. For the newly synthesized proteome data, the number of identified precursors, with consideration of modified

peptide sequences and charge but not SILAC channels, of each protein group was included as a factor for the variance estimation in DEqMS. Gene set enrichment analysis was carried out using the sorted log2 fold change values of all quantified protein groups, using the "GSEA" and "gseGO" function of the clusterProfiler (version 3.18.0)[57] R/Bioconductor package. Gene lists of the Molecular Signatures Database were retrieved and analyzed using the msigdbr (version 7.5.1) *R* package of the CRAN software repository. Protein coding mRNA sequences and codons were derived from the mm10 reference transcript coding sequences with canonical annotation of the Ensembl database. For ribosomal proteins not annotated in these packages, Ensembl transcripts were added. Codon frequencies were calculated for each transcript and averaged across the genome to serve as a reference for the subsets of differentially expressed proteins. The codon usage % deviation from transcriptome average was calculated by subtraction of the reference codon frequency from the codon frequency of an individual transcript. "All proteins" represent all transcript coding sequences (with canonical annotation of the Ensembl database) detected in the newly synthesized or steady-state proteome, respectively. Codon frequencies were calculated individually for AAA, CAA, GAA and the sum of all three codons (VAA). As a control, all codons except the three VAA codons (non-VAA) was used (Suppl. Data 7).

### Statistics & reproducibility

Each experiment was independently repeated at least twice, and statistical analysis was performed using biologically independent replicates. Statistical analyses for proteomics data was carried out using the *R* software environment (version 4.0.3) with software packages specified in the method details section. Samples for LC-MS analysis were acquired in a randomized manner. In other experiments, *p* values were calculated using a two-tailed unpaired *t*-test with Welch correction, two-tailed one sample *t*-test with a hypothetical mean of 1 or two-tailed Wilcoxon rank test, as indicated in the corresponding figure legends. The number of experimental and biologically independent replicates as well as statistical parameters are indicated in the corresponding figure legends. Western blots were quantified using BioRad Image Lab (v6.0.0.25). Analysis of flow cytometry experiments was performed using BD FACSDiva software (v8.0) and FlowJo (v10.6.1). Statistical analyses were performed with the *R* software environment (version 4.0.3) and GraphPad Prism (v10.1.0 (316)).

### Reporting summary

Further information on research design is available in the Nature Portfolio Reporting Summary linked to this article.

## Data availability

The mass spectrometry data generated in this study have been deposited at ProteomeXchange via the PRIDE partner repository with identifiers PXD047316, PXD047935, the RiboSeq and RNA sequencing data at GEO via identifier GSE250593. Source data are provided with this paper.

## Code availability

Scripts for the processing of proteomics, Ribo Seq and RNA Seq data in this paper are available from the lead contact upon request. Any additional information required to reanalyze the data reported in this paper is available from the lead contact upon request.

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

## Acknowledgements

We are grateful to members of the Palm laboratory for experimental advice and critical discussions and Michaela Frye for comments on the manuscript. This work was supported by the Cooperation Program in Cancer Research of the Deutsches Krebsforschungszentrum (DKFZ) and Israel's Ministry of Science and Technology (MOST) (J.H., W.P.), the Cancer Transitional Research and EXchange Program (Cancer-TRAX) within the German-Israeli Helmholtz International Research School (C.P.), the Wilhelm Sander-Stiftung (Projektnummer 2024.086.1 to W.P.) and the European Research Council ('DualRP' ERC StG No. 759579 to F.L.P.).

## Author contributions

J.H. and W.P. conceived and designed research, with contributions from T.B., R.K., A.K., C.P., V.V., F.L.P., J.K., and J.Z. J.H. T.B., R.K., A.K., C.P., V.V., M.S., and D.H. conducted experiments. All authors analyzed and interpreted results. W.P. and J.H. wrote the manuscript with input from all authors. W.P. supervised the study.

## Funding

## Competing interests

J.Z. is a founder, shareholder and scientific advisor of Quantro Therapeutics. J.Z. and the Zuber lab receive research support and funding from Boehringer Ingelheim. Other authors declare no competing interests.
