## [Transparent Peer Review file · Nature Communications]

mTORC1 cooperates with tRNA wobble modification to sustain the protein synthesis machinery

Corresponding Author: Dr Wilhelm Palm

A version of this paper was originally rejected for publication by Nature Communications, however that decision was reconsidered after appeal by the authors.

Version 0:

Reviewer comments:

Reviewer #2

(Remarks to the Author)

In this study by Hermann et al., the authors suggest that the loss of the U34-wobble tRNA modifying enzymes synergizes with mTORC1 inhibition by negatively impacting on new synthesis of ribosomal proteins.

Although the authors perform a set of functional screens, proteomic approaches and ribosome profiling experiments on a variety of cell lines, the conclusions do not add, in my opinion, substantial novelty from the published literature. The importance of the U34-wobble tRNA modifying enzymes in mTORC defective contexts has been widely reported (among others: Candiracci et al, Science advance 2019, Fernández-Vázquez et al., Plos Genetics 2013, Bauer et al., Cell reports 2012, Karlsborn et al. RNA Biol. 2013 [review]). The authors conclude that this dependency is due to the specific inhibition of ribosome protein nascent protein synthesis but this remains purely correlative. The direct impact of the U34-wobble tRNA modifying enzymes on ribosome proteins (i.e.: Rpl29, Rps3) mRNA translation has not been assessed, not supporting the major conclusion of the paper.

Therefore, in my opinion the present study falls short in providing causative demonstration of the proposed mechanism.

Major Concerns:

1. As a general point, statistical analyses (as an example: Fig 1.e; Fig.2f, Fig.2c) are poorly done and not clear making it difficult to evaluate the real effect of the data shown. Importantly, there is a lack of statistic evaluation of the results throughout all the panels shown in the paper. Test used should be put in figure legend.
2. Overexpression of Eip3 and Ctu in iKO cells would be beneficial to support the conclusion of the authors, both on the selective mTORC1 addition to the U34-wobble tRNA modifying enzymes and most importantly on the impact on specific mRNA translation of their suggested targets.
3. Through their CRISPR-Cas9 screen, and subsequent validation in iKO lines, the authors conclude that the U34-wobble tRNA modifying enzymes are "selectively essential during torin 1 treatment" (line 101) and more specifically to mTORC1 activity (line 150-151). In my opinion the data do not fully support the stated conclusions: in sup Fig. 1c the U34-wobble tRNA modifying enzymes show lethal effect upon DMSO treatment compared to control (time 0) independently of torin 1 inhibition. The specificity to mTORC1 is also not clear: pAkt1 levels are not shown upon rapamycin treatment and treatment with Akti show similar cellular growth defect as rapamycin, suggesting that the essential role of the U34-wobble tRNA modifying enzymes is linked to both TORC1 and 2 or more in general cell growth conditions. Synergistic measurement and depletion of RICTOR would help clarifying this point.
4. The data showing the direct impairment of ribosomal proteins mRNA translation do not reflect authors' conclusion. To assess direct translation defects experiments with codon-mutants and direct ribosome density experiments (i.e.: Ribo-IP) should be performed in presence or absence of the U34-wobble tRNA modifying enzymes. This point is key to conclude that the U34-wobble tRNA modifying enzymes directly impact on ribosome protein synthesis
5. Literature is not always well referred in the text: as an example reference 35 is used in lines 322-323 as supporting literature showing that VAA codons do not correlate with decreased protein abundance upon loss of the U34-wobble tRNA modifying enzymes. On the contrary, the referred paper concludes that direct targets of the U34-wobble tRNA modifying enzymes are always enriched of VAA codons but that this is not sufficient for protein downregulation.

Minor comments:

- The use of FC representation for the cell proliferation experiments is not very clear. Would be good to have the effective

number of cells

- Controls for long and short inhibitors treatment are not present
- Labelling of the panels with cell lines used and/or experiment timing would made the reading of the paper easier
- In Fig. 7 the op-puro experiments would be more convincing if done by FACS

Reviewer #3

(Remarks to the Author)

I was asked to evaluate the proteomics data in the paper. There appears to be 2 main experiments. One involving the identification and quantification of nascent proteins using azidohomoalanine (AHA) and SILAC; an approach that was established by one of the authors. The second is a more straightforward quantitative proteomics experiment. The authors have chosen to use a DIA based approach for the identification; employing the DIA-NN software solution. A solution that has had rapid take up by the community although full statistical robustness of the approach wrt identification requires a little more refinement. Statistical analysis is performed using the LIMMA package although the statistical results of that workflow haven't really been used. The authors have essentially reduced the result to a rank-based approach (i.e. top and bottom 10%) of the key nascent proteome data. The volcano and scatter plots show diffused continuous range of ratios across quite a narrow range. Considering the use of 3 replicates and SILAC, the vast majority of proteins in this top/bottom 10% may well be giving the impression of changing due to their variation and not through real change. The data is not presented in a way for the reader to be able to independently judge if a protein is significantly changing. Furthermore, the narrowness of the changes does not reflect the language used by the authors when describing the results. The data should be reformatted to provide more transparency.

Reviewer #4

(Remarks to the Author)

In this manuscript, Hermann et al. studied the functional cooperation between mTORC1, a key regulator of protein synthesis and cell growth, and tRNA wobble uridine-modifying enzymes (U34-enzymes). By performing a proliferation-based CRISPR screen, the authors found that U34-enzymes are essential to sustain cell proliferation upon mTOR inhibition but are dispensable in cells with active mTOR, validating these results both in vitro and in vivo. They found that U34-enzymes, in concert with mTORC1, play a key role in sustaining ribosomal protein synthesis. Accordingly, the combined suppression of mTORC1 and U34-enzymes impaired ribosomal protein synthesis and globally inhibited translation.

This manuscript contains some potentially exciting new findings on the regulation of protein synthesis, by integrating mTORC1 signaling and RNA modification pathways. The experiments are well-designed and the experimental flow is logically linked. However, there are some issues that require further clarification. For instance, the mechanism regulating the crosstalk between mTORC1 and U34-enzymes has not been addressed. These two pathways seem to play synergic roles in the control of protein synthesis, but how they "talk" to each other to properly regulate protein translation is unclear.

Furthermore, when assessing cell proliferation, there is a high variability in the results obtained in different figures using similar experimental conditions, which raises concerns about the robustness of some of the data. Finally, there are issues related to the interpretation of some of the data.

Specific points:

1. The authors claim that "Loss of tRNA wobble enzymes leads to a global decrease in protein synthesis" (lane 153) in untreated cells. However, they also claim that cell proliferation is only marginally affected upon depletion of U34-enzymes. How can such a strong decrease in protein synthesis be irrelevant for cell proliferation? Is cell size affected by Ctu1/Elp3-depletion?
2. Related to point 1, although the authors state that untreated Ctu1/Elp3-KO cells proliferate "slightly more slowly" than control cells, these results show a high degree of variability. For instance, although I agree with the authors that Ctu1/Elp3-depletion only causes a "slight" decrease in cell proliferation in some of the figures (e.g. Figure 2b, 2c and Supplementary Figures 2c, 2f, 2i, and 2j, see DMSO-treated cells), there is a drastically impaired cell proliferation under the same conditions in other figures (e.g. Figure 2a, 1d, 1e, Supplementary Figure 2g, see DMSO-treated cells), raising concerns about the robustness of the data. Importantly, no statistical analyses have been provided for any of these figures, making the interpretation of these data highly arbitrary and thus not meaningful. This is a serious concern that needs to be fixed.
3. Although mTORC1 and U34-enzymes seem to play synergic roles in the control of protein synthesis, the crosstalk between these two pathways has not been addressed. Interestingly, Ctu1- and Elp3-depletion causes a strong and consistent increase in mTORC1 activity (Supplementary Figures 2c and 2e). What is the mechanism leading to mTORC1 hyperactivation in Ctu1- and Elp3-KO cells?
4. To study the cooperation of U34-enzymes and mTORC1 signaling, the authors analyzed cell proliferation/protein synthesis upon pharmacological inhibition of mTORC1 only. In order to analyze the relevance of U34-enzymes under more physiological conditions, the authors should also assess the role of these enzymes on these processes (i.e. cell proliferation/protein synthesis) upon nutrient depletion.
5. On lane 228, the authors state: "However, despite the particularly strong decrease of ribosomal proteins in the newly synthesized proteome, their steady-state levels were only slightly reduced (Fig. 4j, k)". These findings are puzzling. How do the authors interpret these data? Why are the decreased levels of newly synthesized ribosomes not correlated with overall decreased ribosome abundance? Is this related to ribosome degradation (e.g. ribophagy)? And if so, how is this linked with U34-enzymes?

Version 1:

Reviewer comments:

Reviewer #1

(Remarks to the Author)

I would like to thank the authors for their detailed explanations and the additional experiments provided to address my concerns. In my opinion, the revised version of the manuscript is significantly improved, with clearer conclusions now well-supported by the new data and textual revisions.

My initial concern regarding the novelty of the proposed mechanism has been thoroughly addressed, both in the rebuttal letter and through the new experiments included in the manuscript. The paper is easy to read, and the explanations are presented with clarity.

I sincerely appreciate the authors' effort in addressing these points and congratulate them on their excellent work.

Reviewer #2

(Remarks to the Author)

The authors have now provided proteomics data that has been appropriately treated and visualised. From a proteomics perspective, this reviewer no longer has an issue.

Reviewer #3

(Remarks to the Author)

The authors addressed all my previous concerns

Reviewer #1 (Remarks to the Author):

In this study by Hermann et al., the authors suggest that the loss of the U34-wobble tRNA modifying enzymes synergizes with mTORC1 inhibition by negatively impacting on new synthesis of ribosomal proteins.

Although the authors perform a set of functional screens, proteomic approaches and ribosome profiling experiments on a variety of cell lines, the conclusions do not add, in my opinion, substantial novelty from the published literature. The importance of the U34-wobble tRNA modifying enzymes in mTORC defective contexts has been widely reported (among others: Candiracci et al, Science advance 2019, Fernández-Vázquez et al., Plos Genetics 2013, Bauer et al., Cell reports 2012, Karlsborn et al. RNA Biol. 2013 [review]). The authors conclude that this dependency is due to the specific inhibition of ribosome protein nascent protein synthesis but this remains purely correlative. The direct impact of the U34-wobble tRNA modifying enzymes on ribosome proteins (i.e.: Rpl29, Rps3) mRNA translation has not been assessed, not supporting the major conclusion of the paper.

Therefore, in my opinion the present study falls short in providing causative demonstration of the proposed mechanism.

We thank the referee for the thoughtful and constructive criticism, which has helped us to substantially improve our manuscript. In particular, we have conducted additional mechanistic experiments to investigate the role of U₃₄-wobble tRNA modifying enzymes in ribosomal protein synthesis as suggested by the reviewer. Ribo-Seq experiments show that ribosomal protein mRNAs accumulate ribosomes at U₃₄-enzyme-dependent VAA codons. Consistently, replacing VAA codons with the synonymous U₃₄-enzyme-independent VAG codons rescues ribosomal protein expression in Ctu1-deficient cells. These new experiments corroborate our conclusion that the VAA-rich ribosomal protein mRNAs are direct targets of tRNA wobble enzymes (for more details, please see response to *Major Concern 4*).

The reviewer raises concerns regarding the novelty of our manuscript, referring to a series of studies in yeast that established the rapamycin sensitivity of U₃₄-enzyme-deficient cells. To our knowledge, we report for the first time that mammalian cells lacking U₃₄-enzymes are hypersensitive to mTORC1 inhibition. Considering the pronounced differences in mTOR signaling and growth control between unicellular eukaryotes and mammalian cells, we would like to suggest that demonstrating mammalian phenotypes is relevant and novel. However, it is an interesting question whether the mechanisms underlying the rapamycin sensitivity of U₃₄-enzyme-deficient cells are evolutionarily conserved. The following lines of evidence argue otherwise:

The rapamycin sensitivity of U₃₄-enzyme-deficient yeast has been interpreted more broadly as a sensitivity to environmental stresses (similar to the effects of H₂O₂, caffeine, osmotic stress, or detergents). A mechanistic explanation has been provided by defective translation of stress mRNAs (e.g. PMID: 23874237). Work in fission yeast has also linked the rapamycin sensitivity of Elongator-deficient cells to a role of Elongator in promoting the translation of TORC2 components and TORC1 repressors, whose balance regulates sexual differentiation in this organism (PMID: 31223645). In the original manuscript, we conducted GSEA with the different proteomics datasets, which did not identify stress response pathways or mTOR signaling as major phenotypes of mammalian cells lacking U₃₄-enzymes. We now have analyzed the proteomics data in more detail and do not observe any clear changes in proteins that function in stress response or mTOR signaling (Reviewer 1 – Figure 1). Importantly, we have examined a range of growth-suppressive perturbations and environmental stresses and do

not observe increased sensitivity of Ctu1 KO cells – in striking contrast to their exquisite sensitivity to mTORC1 inhibitors (for more details, please see response to *Major Concern 3*).

Reviewer 1 – Figure 1 | Loss of U₃₄-enzymes does not lead to overt changes in stress proteins or mTOR signaling components. Proteome changes in Ctu1 iKO and Elp3 iKO EPP2 cells, quantified by LC-MS (n = 5). **a)** Stress proteins (integrated stress response, ribosome collision sensing); **b)** mTOR signaling components are highlighted.

Although we cannot rule out that loss of U₃₄-enzymes perturbs certain stress response proteins in our cell models, the above results strongly suggest that this is not a dominant phenotype. Rather, we demonstrate that loss of U₃₄-enzymes leads to a biosynthetic defect that converges with mTORC1 inhibition on the depletion of ribosomal proteins. These results identify a novel mechanism for the mTORC1 inhibitor sensitivity in mammalian cells lacking U₃₄-enzymes. In addition to these mechanistic insights, we would like to highlight two other advances of our study that we consider to be highly novel: 1) We report the first nascent proteome of U₃₄-enzyme-deficient cells, which was critical for identifying U₃₄-enzyme-dependent protein groups that eluded previous studies based on traditional steady-state proteomics; 2) We establish biomedical relevance, demonstrating that tumor growth is potently suppressed by rapamycin in combination with Ctu1 KO – which is remarkable because rapamycin by itself has poor efficacy in the vast majority of tumors.

Major Concerns:

1. As a general point, statistical analyses (as an example: Fig 1.e; Fig.2f, Fig.2c) are poorly done and not clear making it difficult to evaluate the real effect of the data shown. Importantly, there is a lack of statistic evaluation of the results throughout all the panels shown in the paper. Test used should be put in figure legend.

We apologize for any shortcomings in the statistical analyses of our data. The reviewer points especially to proliferation data, which we think requires an explanation: We conducted cell proliferation experiments in technical replicates, which to the best of our knowledge is common practice. However, statistical analysis is not meaningful for such data (although sometimes found in the literature) because the measurements are not independent and have very low variance, which would lead to unrealistically low p values. Instead, we sought to address biological significance by investigating different biological contexts, including different gene

knockouts (Ctu1, Elp3), CRISPR approaches (inducible knockout in a pooled population, single-cell derived clonal knockouts), pharmacological inhibitors (torin 1, rapamycin, PI3K/Akt inhibitors) and multiple cell lines.

Nevertheless, we fully agree with the reviewer that statistical analysis for the proliferation experiments is important. To ensure robust and meaningful statistics, we now show key data as replicate means (of technical triplicates) from biologically independent experiments: Ctu1 iKO / Elp3 iKO \pm torin 1 or rapamycin, EPP2 cells (New Fig. 1d, e); clonal Ctu1 KO / Elp3 KO with Ctu1 / Elp3 cDNA rescue \pm rapamycin, EPP2 and KRPC cells (New Fig. 1f – h).

To improve statistics, we have further added the following analyses: p values for all comparisons in the tumor growth experiment (Fig. 2e); improved display and p values for the codon bias in the newly synthesized proteome and steady-state proteome datasets (Fig. 3d, e; 4g, h; Suppl. Fig. 4, 8a, b); independent replicate data points and p values for ribosomal codon occupancy plots (Fig. 3c, Fig. 4d, 5a, b); details on statistical tests in figure legends (except for proteomics and diricore data, whose more complex statistical analyses are described in detail in the methods section).

We respectfully point out that the original manuscript provided statistical analyses for the different proteomics datasets (corresponding to Fig. 3a, b, 4a, i; Suppl. Fig. 8f, g; Suppl. Table 3, 6 in the revised manuscript) as well as for the quantitative Western blotting experiments demonstrating changes in ribosomal protein levels and protein synthesis (corresponding to Fig. 7, 8; Suppl. Fig. 9, 10 in the revised manuscript). Each of these experiments was based on extensive experimentation to obtain biologically independent replicates (nascent proteomics 3 replicates; steady-state proteomics 5 replicates; Western blotting 3 – 8 replicates). For Western blots, p values were shown in the figures and statistical tests stated in the legends. Therefore, we find the reviewer's assessment of 'a lack of statistic evaluation of the results throughout all the panels shown in the paper' a bit harsh.

2. Overexpression of Elp3 and Ctu in iKO cells would be beneficial to support the conclusion of the authors, both on the selective mTORC1 addiction to the U34-wobble tRNA modifying enzymes and most importantly on the impact on specific mRNA translation of their suggested targets.

Thanks for this suggestion. We now have stably reintroduce Ctu1 / Elp3 cDNA into Ctu1 / Elp3 KO cells via lentiviral transduction. Ectopic expression of Ctu1 / Elp3 abrogates the sensitivity of Ctu 1 / Elp 3 KO cells to rapamycin (New Fig. 1f – h) and restores the levels of ribosomal proteins Rpl29 and Rps3 (New Suppl. Fig. 9a, b).

3. Through their CRISPR-Cas9 screen, and subsequent validation in iKO lines, the authors conclude that the U34-wobble tRNA modifying enzymes are “selectively essential during torin 1 treatment” (line 101) and more specifically to mTORC1 activity (line 150-151). In my opinion the data do not fully support the stated conclusions: in sup Fig. 1c the U34-wobble tRNA modifying enzymes show lethal effect upon DMSO treatment compared to control (time 0) independently of torin 1 inhibition.

Gene essentiality is a quantitative and context-dependent trait (see e.g. PMID: 29033457 for a detailed discussion). In the CRISPR screen, we scored genes as selectively essential / non-essential for $|\log_2FC > 2|$, adj. p value < 0.01 for torin 1 versus DMSO. In our experience, these

criteria are sufficiently stringent to capture robust phenotypes. However, we did not mean to imply that selectively essential genes do not have any fitness effects in the control group. Indeed, as the reviewer points out, *Ctu1* KO and *Elp3* KO cells show a moderate decrease in proliferation (of note, they do not show increased lethality). This becomes even more clear in the new proliferation data based on independent experiments (New Fig. 1 d, e). Here, proliferation is decreased in the knockout cells by ~ 15 – 30 %. This effect is somewhat less pronounced in clonal knockout cell lines, conceivably because they adapt over several weeks to the loss of U_{34} -enzymes, but these cells also show overall decreased proliferation (New Fig. 1 f – h). In all experiments, however, mTORC1 inhibition leads to a drastic suppression of proliferation in the knockout cells while only mildly affecting controls. Thus, the hypersensitivity of U_{34} -deficient cells to mTORC1 inhibition is robustly observed across the different experimental models, which is a key conclusion of our study. Thanks for raising this issue; we have revised the manuscript to more accurately describe the proliferation phenotypes that result from loss of U_{34} -enzymes.

The specificity to mTORC1 is also not clear: pAkt1 levels are not shown upon rapamycin treatment and treatment with Akti show similar cellular growth defect as rapamycin, suggesting that the essential role of the U34-wobble tRNA modifying enzymes is linked to both TORC1 and 2 or more in general cell growth conditions. Synergistic measurement and depletion of RICTOR would help clarifying this point.

Thanks for suggesting these experiments to more directly address the function of mTORC1 versus mTORC2. We now provide Western blots for Akt phosphorylation at the mTORC2 site S473, which slightly increases upon rapamycin treatment (New Suppl. Fig. 2h; see also Suppl. Fig. 8c). These data are consistent with previous work showing that rapamycin increases PI3K/mTORC2/Akt signaling due to alleviation of negative feedback (PMID: 17001314, 21659605). Following the reviewer's suggestion, we have also performed genetic experiments to directly address the relevance of mTORC1 versus mTORC2. To this end, we have specifically suppressed Raptor, an mTORC1 component, Rictor, an mTORC2 component, or Rheb, which links mTORC1 activation to upstream Akt signaling. The proliferation of *Ctu1*-deficient cells is strongly suppressed by Raptor or Rheb sgRNAs (New Fig. 2a, c; Suppl. Fig. 3a, c). However, Rictor sgRNA does not have an effect in *Ctu1* KO cells, and Rictor sgRNA-expressing *Ctu1* KO cells have the same rapamycin sensitivity as control cells (New Fig. 2b; Suppl. Fig. 3b). These data confirm that *Ctu1*-deficient cells depend on mTORC1 but not on mTORC2.

The question whether U_{34} -enzymes could be more broadly essential under conditions that affect cell growth is interesting, especially in light of findings in yeast that loss of U_{34} -enzymes renders cells hypersensitive to various stresses. To examine whether this is also the case in mammalian cells, we now have conducted cell proliferation assays using additional perturbations. First, we have suppressed the MAPK pathway, which is the second major pathway besides PI3K/Akt that promotes cell proliferation downstream of growth factor signaling. Increasing concentrations of inhibitors targeting MEK or ERK progressively suppress cell proliferation, to a comparable extent in *Ctu1* KO and control cells (New Suppl. Fig. 3e, f). We have also examined osmotic stress, an environmental stress to which U_{34} -enzyme-deficient yeast is hypersensitive. Adding NaCl to culture medium leads to a concentration-dependent decrease in cell proliferation, which again is comparable for *Ctu1* KO and control cells (New Suppl. Fig. 3g). Taken together, these data strongly argue against a general sensitivity of U_{34} -enzyme-deficient mammalian cells to growth perturbations but rather specifically to the inhibition of mTORC1.

4. The data showing the direct impairment of ribosomal proteins mRNA translation do not reflect authors' conclusion. To assess direct translation defects experiments with codon-mutants and direct ribosome density experiments (i.e.: Ribo-IP) should be performed in presence or absence of the U34-wobble tRNA modifying enzymes. This point is key to conclude that the U34-wobble tRNA modifying enzymes directly impact on ribosome protein synthesis

We agree with the reviewer that codon mutants are a key experiment to demonstrate dependence on U₃₄-enzymes. This experiment is challenging for ribosomal proteins, which are rapidly degraded if not incorporated into ribosomes (PMID: 17446074), making it difficult to study individual proteins in isolation. We now have succeeded by stabilizing ribosomal proteins through proteasome inhibitor treatment. Ribosomal proteins are overall rich in VAA codons (especially AAA), with a few exceptions. To investigate the impact of mRNA codon content on ribosomal protein synthesis, we have generated codon mutants for two ribosomal proteins with VAA-rich mRNAs (Rpl29, Rps25) and one ribosomal protein whose mRNA is not enriched in VAA codons (Rps3). To this end, all VAA codons were replaced with VAG; expression of codon mutant and wild-type proteins (tagged with HA) was analyzed by quantitative Western blotting. Rpl29 and Rps25 levels are significantly decreased in Ctu1 KO cells, whereas the corresponding VAG codon variants are efficiently expressed, at the same level as in control cells (New Fig. 6 c – f). Thus, Rpl29 and Rps25 depend on U₃₄-enzymes for efficient expression in a codon-dependent manner. However, the expression of Rps3, which has a low VAA codon content, does not depend on U₃₄-enzymes (New Fig. 6g, h). Interestingly, the steady-state levels of Rps3 decrease when U₃₄-enzymes and mTORC1 are suppressed (in the absence of proteasome inhibitors) (New Fig. 7). Conceivably, Rps3 is synthesized normally in the absence of U₃₄-enzymes but then degraded, because the perturbed synthesis of other ribosomal proteins prevents its incorporation into ribosomal particles.

To investigate translational defects in U₃₄-enzyme-deficient cells at the mRNA level, we have conducted additional ribosome profiling experiments. Analyzing ribosome occupancy at codon resolution is noisy for individual transcripts, but we obtain sufficient reads if we group ribosomal protein mRNAs together. Indeed, loss of Ctu1 causes a pronounced accumulation of ribosomes at VAA codons of ribosomal protein mRNAs, especially at AAA codons (New Fig. 4d). Together with the high frequency of AAA codons in many ribosomal protein mRNAs, this likely explains why this protein class is particularly dependent on U₃₄-enzymes.

Next, we analyzed the ribosome profiling data to determine ribosomal occupancy at individual transcripts (i.e. across all codons). Torin 1 causes a strong decrease in ribosomal occupancy at ribosomal protein mRNAs (Reviewer 1 – Figure 2a), consistent with previous findings that translational initiation at these TOP motif-containing transcripts is highly dependent on mTORC1. However, loss of Ctu1 does not overtly affect ribosomal occupancy at ribosomal protein mRNAs (Reviewer 1 – Figure 2b). Ribosomal occupancy is also not perturbed at the mRNAs of kinesins (Reviewer 1 – Figure 2c, d), a protein class that was recently identified by the Close lab to be highly dependent on U₃₄-enzymes (PMID: 33859181). This strikingly differs from the results of our nascent proteomics experiments in which loss of Ctu1 or Elp3 led to a strong, consistent decrease in the synthesis of both ribosomal proteins and kinesins (Suppl. Fig. 5c). Thus, loss of U₃₄-enzymes does not affect ribosomal occupancy at established U₃₄-enzymes targets identified by us and others, indicating that the accumulation of ribosomes at VAA codons is either too subtle to be detected at the whole-transcript level or compensated at other steps such as translational initiation. Importantly, ribosomal occupancy does not directly reflect translational output, but rather is interpreted as either decreased or, conversely, increased translational efficiency in different contexts. This highlights the importance of our approach to quantify the translational output of U₃₄-enzyme-deficient cells at the protein level

using nascent proteomics, which was critical to identify ribosomal proteins as targets of this tRNA modification pathway. We thank the reviewer for these detailed experimental suggestions, which have helped us to strengthen our mechanistic conclusion and refine the insights we gain into the relationship between U₃₄-enzymes and codon-dependent synthesis of ribosomal proteins.

Reviewer 1 – Figure 2 | Ribosome profiling detects transcript-level changes in ribosomal occupancy upon mTORC1 inhibition but not Ctu1 deficiency. Transcript-level changes in ribosomal occupancy in EPP2 cells with Ctu1 iKO or torin 1 treatment [300 nM, 2 h], analyzed by ribosome profiling (n = 5 – 6 independent experiments). Ribosomal proteins and kinesins, two U₃₄-enzyme-dependent protein classes, are highlighted.

5. Literature is not always well referred in the text: as an example reference 35 is used in lines 322-323 as supporting literature showing that VAA codons do not correlate with decreased protein abundance upon loss of the U₃₄-wobble tRNA modifying enzymes. On the contrary, the referred paper concludes that direct targets of the U₃₄-wobble tRNA modifying enzymes are always enriched of VAA codons but that this is not sufficient for protein downregulation.

Thanks for pointing out this mistake. When discussing this paper, we meant to reiterate on the author's observation that VAA codon content is surprisingly insufficient to predict protein abundance in U₃₄-enzyme-deficient cells. We have revised the manuscript accordingly and have also improved the manuscript to better discuss previous work including the papers highlighted by the reviewer.

Minor comments:

- *The use of FC representation for the cell proliferation experiments is not very clear. Would be good to have the effective number of cells*

Thanks for pointing this out. To improve the clarity of cell proliferation figures, we have changed the y axis label to ‘Relative cell number (FC)’ – i.e. the relative fold change in cell number that has occurred between the starting and final population – and added an explanation to the figure legend. We suggest that it is helpful to show relative fold change, because this directly indicates how much proliferation has occurred over the course of the experiment. This would not be obvious if effective cell numbers were shown.

- *Controls for long and short inhibitors treatment are not present*

We apologize that the timings of inhibitor treatment were not clear. For cell proliferation, tumor growth, proteomics, ribosome profiling and most Western blotting experiments, the timing of inhibitor and control (DMSO) treatments are matched in a given experiment. The only exception is the time course for torin 1 treatment in Figure 8h, where DMSO is matched to the longest torin 1 time point. To compare knockout and control cells over time, it was important to load all samples on the same gel, which would not have been possible if each torin 1 time point had a matched DMSO control. As torin 1 affects puromycin incorporation only at the longer time points, DMSO controls for the earlier time points are very unlikely to be relevant for the interpretation of this experiment.

- *Labelling of the panels with cell lines used and/or experiment timing would made the reading of the paper easier*

Thanks for this suggestion; we have revised the figures accordingly. If the same cell line was used throughout a figure, we stated this info in the figure legend to avoid redundancy.

- *In Fig. 7 the op-puro experiments would be more convincing if done by FACS*

While Western blotting and flow cytometry are widely used to quantify puromycin incorporation, we have used an orthogonal approach to corroborate the defects in global protein synthesis. To this end, we have developed a nascent proteomics method for absolute quantification of changes in protein synthesis at the proteome scale. This method is based on labeling newly synthesized proteins with the methionine analogue AHA and with SILAC isotopes, followed by enrichment using click chemistry and SILAC quantification. In previous publications, we used this method to compare relative changes in the abundance of labeled proteins between two experimental groups (e.g. PMID: 38086798). In order to quantify absolute changes, we now also measure total, non-enriched proteins in each sample and use this to normalize the nascent proteomics data. Thereby, we reveal a global downshift in protein synthesis in Ctu1 KO and Elp3 KO cells, which is further aggravated by mTORC1 inhibition. These data confirm the results from the puromycin incorporation assays and even allow us to analyze absolute changes in the synthesis of individual proteins.

Reviewer #2 (Remarks to the Author):

I was asked to evaluate the proteomics data in the paper. There appears to be 2 main experiments. One involving the identification and quantification of nascent proteins using azidohomoalanine (AHA) and SILAC; an approach that was established by one of the authors. The second is a more straightforward quantitative proteomics experiment. The authors have chosen to use a DIA based approach for the identification; employing the DIA-NN software solution. A solution that has had rapid take up by the community although full statistical robustness of the approach wrt identification requires a little more refinement. Statistical analysis is performed using the LIMMA package although the statistical results of that workflow haven't really been used. The authors have essentially reduced the result to a rank-based approach (i.e. top and bottom 10%) of the key nascent proteome data. The volcano and scatter plots show diffused continuous range of ratios across quite a narrow range. Considering the use of 3 replicates and SILAC, the vast majority of proteins in this top/bottom 10% may well be giving the impression of changing due to their variation and not through real change. The data is not presented in a way for the reader to be able to independently judge if a protein is significantly changing. Furthermore, the narrowness of the changes does not reflect the language used by the authors when describing the results. The data should be reformatted to provide more transparency.

We thank the reviewer for the detailed suggestions concerning the analysis and presentation of the proteomics data. In our manuscript, we investigate tRNA U₃₄-enzymes, which generate a class of tRNA modifications that ensures efficient translation of VAA codons. We and others observe that the loss of these tRNA modifications has a moderate impact on the synthesis of a large range of proteins (the present study; PMID: 33859181), a phenotype that indeed is challenging to study with proteomics approaches. DIA-based proteomics has gained significant traction in the field by enabling deep proteome analysis, and we would like to emphasize that DIA-NN has been thoroughly benchmarked with regard to false discoveries and quantitative accuracy (e.g. PMID: 36609502; PMID: 35551187). In addition, we have benchmarked DIA-NN specifically in the context of nascent proteome analysis via QuaNPA (PMID 38086798), i.e. the approach used in this manuscript. To address biological robustness, instead of increasing the number of experimental replicates we examined two different tRNA U₃₄-enzyme knockouts (Ctu1, Elp3) and observed consistent phenotypes in their newly synthesized and steady state proteomes (Fig. 4b, c, i, k; Suppl. Fig. 5c, 6). In addition, we examined the reciprocal effect of Ctu1 knockout and two distinct mTORC1 inhibitors (rapamycin, torin 1) on ribosomal protein synthesis, observing a high degree of correlation for the changes of individual ribosomal proteins between the different proteomics datasets (Fig. 6a, b; Suppl. Fig. 8f, g). Based on the reviewer's suggestions, we have taken the following additional measures to ensure that the reported changes reflect real biological differences rather than experimental variation:

- To directly compare changes of individual proteins between Ctu1 and Elp3 knockout cells, we now have generated correlation plots for the newly synthesized proteome and steady-state proteome (New Suppl. Fig. 5a, 7). The proteome changes of the two knockouts are well correlated, including the decrease in ribosomal proteins, consistent with Ctu1 and Elp3 functioning in the same tRNA modification pathway.
- In the original manuscript, we used a rank-based approach to analyze the mRNA codon content of the 10 % most decreased/increased proteins in the newly synthesized proteome, as these are most likely to be dependent on/independent of Ctu1 and Elp3. To improve data representation, we now show the entire dataset in decile ranks of log₂FC (i.e. 0-10%, 10-20%, 20-30% ...). Both knockouts show a significant,

progressive increase in mRNA VAA codon content towards the lower deciles (i.e. decrease in the newly synthesized proteome / steady state proteome) (New Fig. 3d, e; New Suppl. Fig. 4b). Conversely, VAA codon content decreases towards the higher deciles (i.e. increase in the newly synthesized proteome / steady state proteome). The reciprocal changes are observed for non-VAA codons, whose efficient translation does not depend on Ctu1 and Elp3.

- In addition to the rank-based approach, we provide correlation plots comparing change in abundance and mRNA codon content for individual proteins. The change of proteins in the newly synthesized proteome of Ctu1 and Elp3 knockout cells is negatively correlated with VAA codon content and, conversely, positively correlated with non-VAA codon content (New Suppl. Fig. 4a).
- We have conducted an additional nascent proteomics experiment in Ctu1 knockout cells (in 3 technical replicates). This independent dataset reproduces the results of the first experiment, confirming the key conclusions of our manuscript: 1) the synthesis of most proteins is decreased upon loss of Ctu1; 2) VAA codons are significantly enriched in mRNAs of the most depleted proteins, whereas non-VAA codons are depleted; 3) GSEA identifies ribosomal proteins and other proteins functioning in RNA-related processes as the most decreased protein classes, extracellular matrix proteins as the most increased protein class (Reviewer 2 – Figure 1).

Taken together, we observe consistent phenotypes for two different tRNA U_{34} -enzyme knockouts and in independent proteomics experiments. Collectively, these gradual trends strongly indicate a Ctu1/Elp3-dependent change in codon representation, and they argue against (random) variation. Moreover, we use quantitative Western blotting as an orthogonal approach to investigate multiple cell lines, different tRNA U_{34} -enzyme knockouts and mTORC1 inhibitors. These experiments fully support the key conclusions of our proteomics experiments – depletion of ribosomal proteins and reduced protein synthesis upon suppression of U_{34} enzymes and mTORC1 (Fig. 7, 8; Suppl. Fig. 9c, d, 10). Of note, while the effect size in the proteomics experiments may appear small, ribosomal proteins account for ~ 5 % of total cellular protein, which translates to a substantial change in a cell's biosynthetic output.

Following the reviewer's suggestions, we have further revised the manuscript to improve the reporting, transparency and discussion of the proteomics results as follows:

- In the methods section, we detail the steps that were taken to ensure reproducible and reliable MS measurements including wash runs after each sample analysis to avoid sample carryover; regular QC runs to ensure high and consistent MS performance; and block randomization (PMID: 32969222) of all samples for LFQ experiments.
- We have added the complete dataset for VAA codon content (New Supp. Table 7) and tables with the LFC, adjusted p value and VAA codon content of the different protein subgroups shown in Figures 4e, f and Suppl. Fig. 5c (New Suppl. Table 5).
- We have generated coefficient of variation plots for nascent proteomics experiments, which indicate high reproducibility between replicates. Consistently, PCA demonstrates that the technical replicates of each experimental group cluster closely together (Reviewer 2 – Figure 2).

- We have revised the manuscript to more accurately describe the changes observed in the different proteomics experiments.

Reviewer 2 – Figure 1 | A biologically independent nascent proteomics experiment confirms the importance of Ctut1 for ribosomal protein synthesis. New, independent analysis of the newly synthesized proteome in Ctut1 iKO EPP2 cells (experiment was conducted analogous to Figure 3a, d; Figure 4a; Suppl. Figure 4b). **a)** Changes in the newly synthesized proteome, analyzed with the quantitative analysis of the newly synthesized proteome (QuaNPA) workflow (n = 3 experimental replicates). **b), c)** Correlation of b) VAA, c) non-VAA codon deviation and protein synthesis changes. Protein changes are shown as decile rank of the \log_2FC (i.e. 1 = 10 % lowest \log_2FC ; 10 = 10 % highest \log_2FC) in the newly synthesized proteome. Box plots represent 10th to 90th percentiles and median. Grey line indicates codon usage median of all quantified proteins. p values were calculated by Wilcoxon rank test. **d)** GSEA showing the most depleted and enriched protein groups. NES, normalized enrichment score.

Reviewer 2 – Figure 2 | Nascent proteomics measurements show high consistency across replicates. **a)** Principle component analysis of three experimental replicates quantified for the newly synthesized proteome of Ctu1 iKO and Elp3 iKO EPP2 cells \pm mTORC1 inhibitors, using the mass spectrometry based quantitative analysis of the newly synthesized proteome (QuaNPA) workflow. **b)** Coefficient of variation for the newly synthesized proteome datasets of Ctu1 iKO and Elp3 iKO EPP2 cells. Ribosomal proteins are highlighted.

Reviewer #3 (Remarks to the Author):

In this manuscript, Hermann et al. studied the functional cooperation between mTORC1, a key regulator of protein synthesis and cell growth, and tRNA wobble uridine-modifying enzymes (U34-enzymes). By performing a proliferation-based CRISPR screen, the authors found that U34-enzymes are essential to sustain cell proliferation upon mTOR inhibition but are dispensable in cells with active mTOR, validating these results both in vitro and in vivo. They found that U34-enzymes, in concert with mTORC1, play a key role in sustaining ribosomal protein synthesis. Accordingly, the combined suppression of mTORC1 and U34-enzymes impaired ribosomal protein synthesis and globally inhibited translation.

This manuscript contains some potentially exciting new findings on the regulation of protein synthesis, by integrating mTORC1 signaling and RNA modification pathways. The experiments are well-designed and the experimental flow is logically linked. However, there are some issues that require further clarification. For instance, the mechanism regulating the crosstalk between mTORC1 and U34-enzymes has not been addressed. These two pathways seem to play synergic roles in the control of protein synthesis, but how they “talk” to each other to properly regulate protein translation is unclear. Furthermore, when assessing cell proliferation, there is a high variability in the results obtained in different figures using similar experimental conditions, which raises concerns about the robustness of some of the data. Finally, there are issues related to the interpretation of some of the data.

We thank the reviewer for evaluating our study as potentially exciting, with well-designed, logically linked experiments, and appreciate the thoughtful suggestions that have helped to improve the robustness and conclusiveness of our data.

Specific points:

1. The authors claim that “Loss of tRNA wobble enzymes leads to a global decrease in protein synthesis” (lane 153) in untreated cells. However, they also claim that cell proliferation is only marginally affected upon depletion of U34-enzymes. How can such a strong decrease in protein synthesis be irrelevant for cell proliferation? Is cell size affected by Ctu1/Elp3-depletion?

Thanks for pointing out this paradox. As suggested by the reviewer, we have quantified cell size, which is comparable between Ctu1 KO / Elp3 KO and control cells (New Suppl. Fig. 2c). We have also quantified protein content / cell, which is not significantly changed in U₃₄-enzyme-deficient cells but shows a slightly decreased trend (New Suppl. Fig. 2d). With regard to cell proliferation, in the original manuscript we described the proliferative defects of Ctu1 KO and Elp3 KO cells as slight to moderate in different contexts. We now have performed additional experiments to investigate these phenotypes in more detail (please also see response to *Specific Point 2*), confirming a moderate decrease in cell proliferation upon loss of U₃₄-enzymes. We do not think that the observed protein synthesis and growth defects of U₃₄-enzyme-deficient cells are at odds, because it is unclear how precisely to relate the effect size of the different measurements, i.e. proteomics and cell proliferation. However, both proliferation and protein synthesis are decreased in U₃₄-enzyme deficient cells, and treating cells with mTORC1 inhibitors consistently leads to more severe defects. We have revised the manuscript to clarify this point.

2. Related to point 1, although the authors state that untreated Ctu1/Elp3-KO cells proliferate “slightly more slowly” than control cells, these results show a high degree of variability. For instance, although I agree with the authors that Ctu1/Elp3-depletion only causes a “slight” decrease in cell proliferation in some of the figures (e.g. Figure 2b, 2c and Supplementary Figures 2c, 2f, 2i, and 2j, see DMSO-treated cells), there is a drastically impaired cell proliferation under the same conditions in other figures (e.g. Figure 2a, 1d, 1e, Supplementary Figure 2g, see DMSO-treated cells), raising concerns about the robustness of the data. Importantly, no statistical analyses have been provided for any of these figures, making the interpretation of these data highly arbitrary and thus not meaningful. This is a serious concern that needs to be fixed.

We apologize for any shortcomings in the statistical analysis and interpretation of the cell proliferation data. We think this requires an explanation: In the original manuscript, we conducted cell proliferation experiments in technical replicates, which to the best of our knowledge is common practice. However, statistical analysis is not meaningful for such data (although sometimes found in the literature) because the measurements are not independent and have very low variance, which would lead to unrealistically low p values. Instead, we sought to address biological significance by investigating different biological contexts, including different gene knockouts (Ctu1, Elp3), CRISPR approaches (inducible knockout in a pooled population, single-cell derived clonal knockouts), pharmacological inhibitors (torin 1, rapamycin, PI3K/Akt inhibitors) and multiple cell lines. The impact of Ctu1 KO or Elp3 KO on cell proliferation indeed showed some variability, which we described as slight to moderate.

Nevertheless, we fully agree with the reviewer that statistical analysis for the proliferation experiments is important. To ensure robust and meaningful statistics, we now show key data as replicate means (of technical triplicates) from biologically independent experiments: Ctu1 iKO / Elp3 iKO \pm torin 1 or rapamycin, EPP2 cells (New Fig. 1d, e). These data confirm that loss of Ctu1 or Elp3 leads to a significant decrease in cell proliferation, by $\sim 15 - 30\%$. We also have plotted replicate means for clonal knockout cell lines: Ctu1 KO / Elp3 KO with Ctu1 / Elp3 cDNA rescue \pm rapamycin, EPP2 and KRPC cells (New Fig. 1 f – h). Here, the decrease in cell proliferation is somewhat less pronounced, conceivably because the knockout clones adapt over several weeks to the loss of U₃₄-enzymes, but these cells also show overall decreased proliferation (New Fig. 1 f, g). Importantly, in all experiments mTORC1 inhibition leads to a drastic, highly significant suppression of proliferation in the knockout cells while only mildly affecting controls. Thus, the hypersensitivity of U₃₄-deficient cells to mTORC1 inhibition is robustly observed across the different experimental models, which is a key conclusion of our study. We thank the reviewer for raising this issue, which has helped us to more accurately characterize the phenotypes of U₃₄-enzyme-deficient cells.

3. Although mTORC1 and U34-enzymes seem to play synergic roles in the control of protein synthesis, the crosstalk between these two pathways has not been addressed. Interestingly, Ctu1- and Elp3-depletion causes a strong and consistent increase in mTORC1 activity (Supplementary Figures 2c and 2e). What is the mechanism leading to mTORC1 hyperactivation in Ctu1- and Elp3-KO cells?

While our study reveals that mTORC1 and U₃₄-enzymes converge on ribosomal protein synthesis, it is an interesting question whether the two pathways also regulate each other directly. In the original manuscript, we reported that mTORC1 inhibition does not affect the translation of VAA codons or aggravate the accumulation of ribosomes at VAA codons in U₃₄-enzyme-deficient cells (Fig. 5 a, b). To explore the reverse possibility, i.e. regulation of the

mTORC1 signaling pathway by U₃₄-enzymes, first we have reanalyzed our proteomics datasets. Loss of U₃₄-enzymes does not lead to any consistent changes in the abundance of mTOR pathway components (Reviewer 3 – Figure 1a). As the reviewer points out, U₃₄-enzyme-deficient cells sometimes show increased mTORC1 activity as compared to controls. However, this phenotype is variable in our experiments and does not correlate with the the hypersensitivity of U₃₄-enzyme-deficient cells to mTORC1 inhibitors, which we robustly observe across different experiments. To investigate this in more detail, we now have conducted quantitative Western blotting in Ctu1 KO and Elp3 KO cells in multiple biologically independent replicates. Under steady-state conditions, mTORC1 signaling is comparable between Ctu1 KO / Elp3 KO cells and controls in most replicates; however, the knockout cells indeed show higher mTORC1 signaling in some instances (Reviewer 3 – Figure 1b, c). Because mTORC1 activity is controlled by environmental factors that can change in cell culture over time, we also examined acute activation of mTORC1 signaling by insulin or amino acids. Here, Ctu1 KO / Elp3 KO cells and controls show comparable, consistent levels of mTORC1 activity (Reviewer 3 – Figure 1b – d). We suspect that mTORC1 activity displays some variability under steady-state culture conditions due to changes in the environment (e.g. depletion of growth factors or nutrients) or secondary effects that could result e.g. from metabolic differences between control and knockout cells.

Overall, these data argue against a function of U₃₄-enzymes as critical regulators of mTORC1 signaling, and vice versa. Rather, our data are consistent with a model that mTORC1 and U₃₄-enzymes converge on promoting the synthesis of ribosomal proteins, which are unique in being encoded by mRNAs that have a high AAA codon content and are under tight control by mTORC1. We cannot rule out that cross-regulation between mTORC1 and U₃₄-enzymes occurs in certain contexts, which remains an intriguing possibility, but to explore this further is beyond the scope of the present manuscript.

Reviewer 3 – Figure 1 | U₃₄-enzymes are not required for mTORC1 signaling. **a)** Proteome changes in Ctu1 iKO and Elp3 iKO EPP2 cells, quantified by LC-MS (n = 5). mTOR signaling components are highlighted. **b)** mTORC1 signaling activity in Ctu1 iKO and Elp3 iKO EPP2 cells under steady state conditions and upon insulin stimulation, analyzed by immunoblotting. 4 independent replicates are shown. **c)** Quantification of the immunoblotting data shown in b). Data are represented as mean ± SEM (n = 4 independent experiments). p values were calculated by two-sided unpaired *t*-test with Welch correction. **d)** mTORC1 signaling activity in Ctu1 iKO EPP2 cells upon amino acid withdrawal and amino acid restimulation, analyzed by immunoblotting.

4. To study the cooperation of U34-enzymes and mTORC1 signaling, the authors analyzed cell proliferation/protein synthesis upon pharmacological inhibition of mTORC1 only. In order to analyze the relevance of U34-enzymes under more physiological conditions, the authors should also assess the role of these enzymes on these processes (i.e. cell proliferation/protein synthesis) upon nutrient depletion.

Thanks for suggesting these experiments. To corroborate the findings based on experiments with pharmacological mTORC1 inhibitors, first we have conducted a series of genetic experiments. To this end, we have genetically suppressed Raptor, an mTORC1-specific subunit, or Rheb, an upstream activator of mTORC1, using CRISPR/Cas9. Raptor depletion strongly suppresses the proliferation of Ctu1 KO cells, while only moderately affecting control cells. Similarly, Rheb depletion selectively decreases the proliferation of Ctu1 KO cells (Reviewer 3 – Figure 2a, b; New Fig. 2a – c; Suppl. Fig. 3a – c). These experiments provide genetic evidence that loss of U34-enzymes sensitizes cells to the suppression of mTORC1.

Reviewer 3 – Figure 2 | U34-enzyme-deficient cells show increased sensitivity to inactivation of mTORC1. a), b) Fold change in cell number of Ctu1 KO and control EPP2 cells \pm a) Raptor iKO, b) Rheb iKO after 3 days of culture. c) Fold change in cell number of Ctu1 iKO and control EPP2 cells after 3 days of culture in media containing different FBS concentrations. d) Puromycin (puro) incorporation assay in Ctu1 iKO EPP2 cells after 16 h culture in media containing different FBS concentrations, analyzed by immunoblotting. a) – c) Data are represented as mean \pm SD (n = 3 technical replicates).

To decrease mTORC1 signaling in a more physiological manner, we have deprived cells of growth factors by reducing FBS in the medium. Ctu1 KO cells display increased sensitivity to serum starvation, both with regard to proliferation and protein synthesis (Reviewer 3 – Figure 2c, d). The effects are less pronounced than those we observe for genetic / pharmacological mTORC1 inhibition, consistent with residual mTORC1 activity in serum-starved cells. Nevertheless, these results corroborate our conclusions concerning the increased sensitivity of U₃₄-enzyme-deficient cells to suppression of the PI3-kinase/Akt/Rheb signaling axis, which communicates growth factor signals to mTORC1. Following the reviewer's suggestion, we have also deprived cells of various combinations of amino acids, which are the key nutrient input into mTORC1 (e.g. all amino acids, essential amino acids, sulfur-containing amino acids [which provide the substrate for Ctu-mediated tRNA thiolation]). We observe a comparable decrease proliferation and translation in Ctu1 KO and control cells in response to these perturbations. In our experience, amino acid levels have to be lowered drastically in cell culture media to inactivate mTORC1, which at the same time deprives cells of the building blocks required to make proteins. Conceivably, the dual function of amino acids in promoting pro-translation signaling and serving as biosynthetic substrates masks the impact of Ctu1 KO on protein synthesis, which might explain why we only observe effects when suppressing growth factor signaling inputs into mTORC1.

5. On lane 228, the authors state: "However, despite the particularly strong decrease of ribosomal proteins in the newly synthesized proteome, their steady-state levels were only slightly reduced (Fig. 4j, k)". These findings are puzzling. How do the authors interpret these data? Why are the decreased levels of newly synthesized ribosomes not correlated with overall decreased ribosome abundance? Is this related to ribosome degradation (e.g. ribophagy)? And if so, how is this linked with U₃₄-enzymes?

Thanks for raising this point – in principle, increased ribophagy could contribute to the decrease in ribosomal proteins, especially in the context of mTORC1 inhibition and the ensuing induction of autophagy. To address this, first we have examined whether loss of U₃₄-enzymes affects autophagy. Autophagic flux is comparable in Ctu1 KO / Elp3 KO cells and controls (New Suppl. Fig. 9e). To determine whether autophagy was required for the decreased ribosomal protein levels, next we have deleted the essential autophagy component Atg5. The combination of mTORC1 inhibition and Ctu1 KO strongly decreases ribosomal protein levels even in Atg5-deficient cells (New Suppl. Fig. 9f). Thus, autophagy is not required for the depletion of ribosomal proteins that results from co-suppression of mTORC1 and U₃₄-enzymes.

We fully agree with the reviewer that the discrepancy between the decreased synthesis of ribosomal proteins and the less pronounced reduction of their steady-state levels upon loss of U₃₄-enzymes is intriguing. In fact, we consider this a key finding of our work. This phenotype was not observed in previous studies that characterized U₃₄-enzyme-deficient cells using traditional steady-state proteomics (PMID: 23935536, 33859181), reporting a surprisingly mild impact on the proteome – despite transcriptome-wide perturbations in VAA codon translation. We were only able to systematically identify U₃₄-enzyme-dependent proteins by developing a nascent proteomics method to quantify changes in protein synthesis directly (New Fig. 3a, b). Importantly, a decreased rate of protein synthesis is expected to translate to a decrease in steady protein levels if the rate of cell growth remains constant. This is clearly not the case for U₃₄-enzyme-deficient cells. Our results suggest that in this context, cells prevent the dilution of ribosomal proteins (and other U₃₄-enzyme-dependent proteins) through adaptive mechanisms, which likely involve a reduction in their growth rate. We cannot exclude the possibility that other homeostatic mechanisms contribute, but this will be subject of future studies.